# Modelling human neuronal catecholaminergic pigmentation in rodents recapitulates age-related neurodegenerative deficits

Ariadna Laguna [1,2,3,11], Núria Peñuelas [1,2,11], Marta Gonzalez-Sepulveda [1,2], Alba Nicolau[1,2], Sébastien Arthaud [4,5], Camille Guillard-Sirieix [1,2], Marina Lorente-Picón [1,2], Joan Compte [1,2], Lluís Miquel-Rio [6,7], Helena Xicoy[1], Jiong Liu[4,5], Annabelle Parent [1,2], Thais Cuadros [1,2], Jordi Romero-Giménez [1], Gemma Pujol[1], Lydia Giménez-Llort [3,8], Patrice Fort [4,5], Analia Bortolozzi [6,7], Iria Carballo-Carbajal[1] & Miquel Vila [1,2,3,9,10] ✉

One key limitation in developing effective treatments for neurodegenerative diseases is the lack of models accurately mimicking the complex physio-pathology of the human disease. Humans accumulate with age the pigment neuromelanin inside neurons that synthesize catecholamines. Neurons reaching the highest neuromelanin levels preferentially degenerate in Parkinson's, Alzheimer's and apparently healthy aging individuals. However, this brain pigment is not taken into consideration in current animal models because common laboratory species, such as rodents, do not produce neuromelanin. Here we generate a tissue-specific transgenic mouse, termed tgNM, that mimics the human age-dependent brain-wide distribution of neuromelanin within catecholaminergic regions, based on the constitutive catecholamine-specific expression of human melanin-producing enzyme tyrosinase. We show that, in parallel to progressive human-like neuromelanin pigmentation, these animals display age-related neuronal dysfunction and degeneration affecting numerous brain circuits and body tissues, linked to motor and non-motor deficits, reminiscent of early neurodegenerative stages. This model could help explore new research avenues in brain aging and neurodegeneration.

Humans progressively accumulate with age the dark-brown pigment neuromelanin (NM) within catecholaminergic brain nuclei. NM is formed as a byproduct of catecholamine synthesis and metabolism, beginning with the conversion of tyrosine to L-DOPA, which is then synthesized further to form the neurotransmitters dopamine and noradrenaline, the cytosolic excess of which, if not encapsulated into synaptic vesicles, is oxidized into NM[1]. In contrast to the widespread distribution of other brain pigments such as lipofuscin, NM is thus restricted to catecholamine-producing regions and forms only in neurons. Because neurons do not have the capacity to degrade or eliminate this pigment, NM progressively accumulates with age until occupying most of the neuronal cytoplasm[2]. It has long been

established that neurons with the highest NM levels, including midbrain dopaminergic neurons from the substantia nigra pars compacta (SNpc/A9) and ventral tegmental area (VTA/A10) and pontine noradrenergic neurons from the locus coeruleus (LC/A6), preferentially degenerate in Parkinson's disease (PD) leading to characteristic motor and non-motor symptoms[3]. In addition, the LC also degenerates extensively in patients with Alzheimer's disease (AD)[4]. In both disorders, the LC is in fact one of the first nuclei to become affected, underlying prodromal symptoms such as sleep disorders, anxiety or depression[5–7]. Even in the absence of overt PD or AD, NM-filled neurons from apparently healthy aged individuals also exhibit early signs of neuronal dysfunction and degeneration compared to young adult brains, including age-related loss of pigmented SNpc and LC neurons, downregulation of phenotypic neuronal markers, alpha-synuclein and tau pathology or the presence of extracellular NM (eNM) released from dying neurons associated with sustained microglial activation[8–12].

While various factors may contribute to the preferential susceptibility of melanized neurons in neurodegenerative disease and brain aging, including their long and diffuse axonal projections, intrinsic pacemaker activity and high metabolic demand[3], the presence of conspicuous intracellular pigment deposits has long been suspected as a key vulnerability factor contributing to their early and extensive demise[13]. However, the potential influence of age-dependent NM accumulation on neuronal function and viability has been largely overlooked in experimental animal modeling because, in contrast to humans, common laboratory animal species such as rodents lack this pigment. Although NM is present in some other species as varied as monkeys[14,15], dolphins[16] and frogs[17], the highly abundant amount of NM in the brainstem, visible with the naked eye, is unique to humans, as NM accumulation increases progressively as the evolutionary relation to humans becomes closer[15]. Thus, most of what is known about the formation and potential role of this pigment in both health and disease is inferred from human postmortem analyses[18].

To overcome this major limitation, here we generated a tissue-specific transgenic mouse (termed tgNM) that mimics the progressive, age-dependent brain-wide distribution of pigmentation within all catecholaminergic regions of the human brain. This model is based on the constitutive catecholaminergic-specific expression of the human melanin-producing enzyme tyrosinase (TYR) driven by the tyrosine hydroxylase (TH) promoter. In a previous proof-of-concept study we have shown that unilateral viral vector-mediated expression of TYR restricted to the rodent SNpc induced the production of a NM-like pigment within ipsilateral nigral dopaminergic neurons virtually analogous to human NM, at levels up to those reached in elderly humans[19]. In the latter animals, progressive NM build-up in the SNpc ultimately compromised neuronal function and triggered nigrostriatal degeneration associated to motor deficits[19,20]. In the tgNM mice reported here, we found that progressive brain-wide bilateral NM pigmentation is associated to age-related neuronal dysfunction and degeneration affecting numerous brain neurotransmission systems and tissues in the body linked to a myriad of motor and non-motor alterations, equivalent to prodromal/early stages of neurodegeneration.

## Results

### Brain-wide and age-dependent human-like NM accumulation in tgNM mice

Constitutive and catecholaminergic-specific overexpression of TYR was achieved by classical transgenesis driven by the TH promoter (Tg(Th-TYR) mice; tgNM). TYR cDNA expression was assessed in distinct catecholaminergic and non-catecholaminergic dissected brain nuclei from tgNM mice at 12 months (m) of age (Fig. S1A). As expected, higher TYR expression was detected within catecholaminergic brain areas compared with non-catecholaminergic regions, with a positive correlation between TYR and TH expression levels (Fig. S1A, B). TYR expression was also confirmed at the cellular level by in situ

hybridization in tgNM brain tissue sections (Fig. S1C). As anticipated from our previous work[19], TYR overexpression resulted in high NM levels in the most PD-vulnerable catecholaminergic regions (SNpc, VTA and LC), appearing as macroscopically visible dark-brown colored areas in the absence of any staining, mimicking their appearance in human specimens (Fig. 1A, C and Movie S1). As in humans, melanized SNpc, VTA and LC from tgNM mice could also be detected macroscopically as hyperintense areas by NM-sensitive high-resolution T1-weighted magnetic resonance imaging (Fig. 1A, C). Histologically, NM from tgNM mice appeared as a dark-brown neuronal pigment in hematoxylin and eosin-stained sections and stained prominently with the melanin marker Masson-Fontana, similar to melanized human neurons (Fig. 1A, C). NM production in tgNM mice was not limited to these major catecholaminergic nuclei but was also present, at varying degrees, in all catecholaminergic cell groups (i.e., A1-A16) (Fig. 2A and Table S1), thereby mimicking the pattern of human NM distribution[21]. As in humans[22], NM is continuously produced throughout life in tgNM mice and progressively accumulates with age until occupying a major portion of the neuronal cytoplasm (Fig. S2A-B). In these animals, NM exhibits a caudorostral gradient of accumulation, reaching earlier higher intracellular levels in lower brainstem areas (dorsal vagal complex [DVC] and LC) than in upper brainstem regions (SNpc and VTA) (Fig. 2B). These results indicate that TYR expression in tgNM mice leads to the production of human-like NM within all brain catecholaminergic neuronal groups in an age-dependent manner.

### Dopaminergic dysfunction and pathology in aged tgNM mice

SNpc and adjacent VTA constitute the main catecholaminergic nuclei in the midbrain and are the primary source of NM in the human brain. Both dopaminergic nuclei degenerate in PD[23]. We assessed the consequences of intracellular NM build-up in terms of viability and function of dopaminergic neurons (Fig. S3A). We characterized first the sensorimotor capacities of these animals at different ages (i.e., young [3–4 m], adult [8–12 m], old [18–20 m]), corresponding to progressively increased levels of NM in SNpc and VTA (Fig. S2). Adult and old tgNM mice spent more time to cross a horizontal beam (Fig. 3A), indicating impaired motor coordination and balance. Adult and old tgNM mice also exhibited decreased olfactory acuity to discriminate a lemon essence (Fig. 3B), indicating impaired olfaction. We also noticed a significant absence of vocalizations during the performance of the behavioral tests that was already present in young tgNM mice (Fig. 3C), a phenotype related to cranial sensorimotor deficits previously reported in alpha-synuclein-overexpressing transgenic mice as an early pre-motor manifestation[24]. Other behavioral tests, such as grip strength and novel object recognition, did not reveal any differences between genotypes at any age (Fig. S3B, C). We then performed stereological cell counts of TH-positive neurons in tgNM mice and noted a significant decrease both in SNpc and VTA at adult and old ages (Fig. 3D, E). This decrease appeared to correspond to a TH phenotypic downregulation rather than cell loss, as indicated by the presence of dopaminergic NM-containing neurons immunonegative for TH (Fig. 3F). These TH-immunonegative pigmented neurons, which are also conspicuously observed in human PD and aged postmortem brains[25,26], represent dysfunctional dopaminergic neurons at early stages of degeneration[13,19]. The loss of dopaminergic phenotype was confirmed by decreased transcript expression of dopaminergic markers in SN-VTA dissected tissue from old tgNM mice (Fig. S4A). To further assess cell death, we performed stereological cell counts of total dopaminergic NM-containing neurons, including TH-immunopositive and TH-immunonegative neurons, which confirmed a lack of significant cell loss in the SNpc and VTA of tgNM mice (Fig. 3G). However, tgNM brain sections showed considerable amounts of extraneuronal NM (eNM) granules, especially at old ages (Fig. S4B). The presence of eNM, which derives from dying neurons and is typically observed in PD and aged postmortem brains[8,25,27,28], indicates an

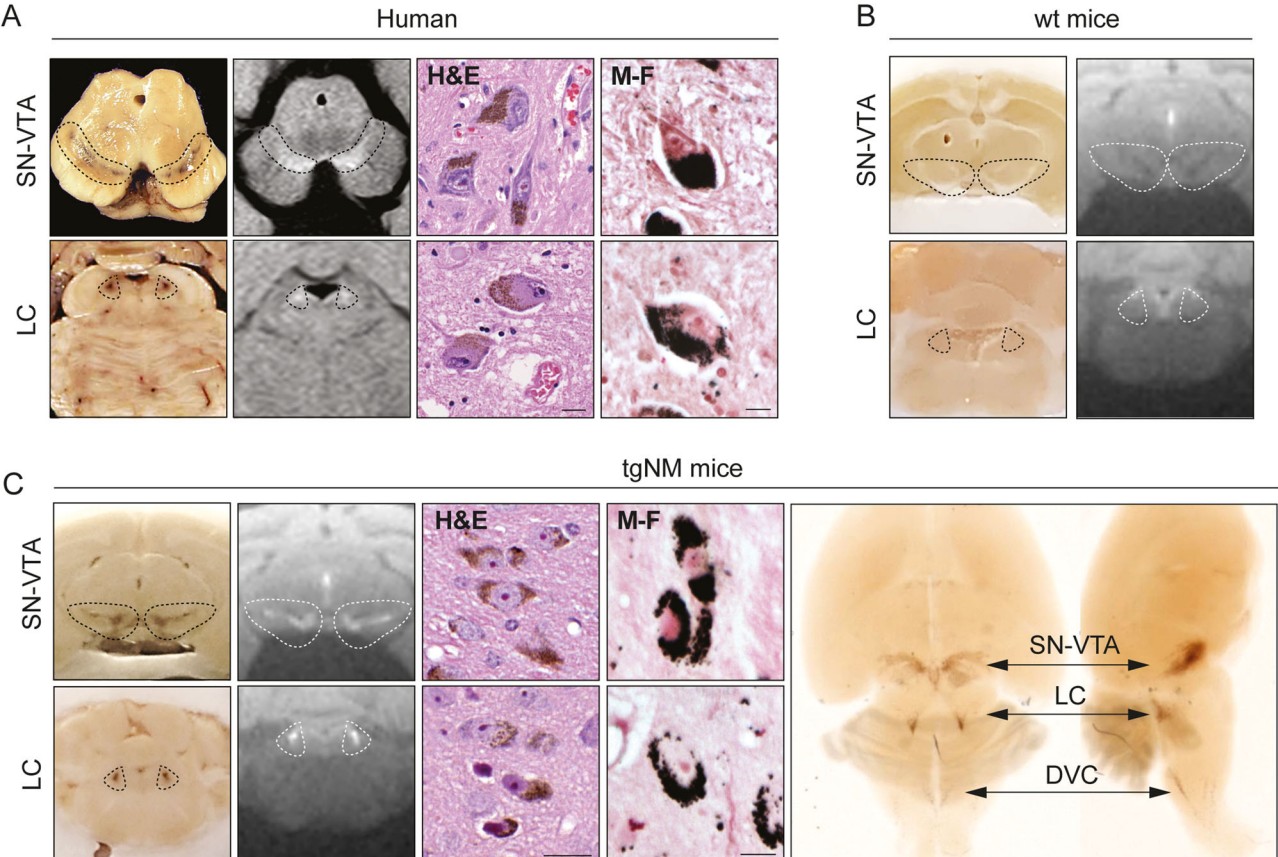

**Fig. 1 | Accumulation of human-like NM in PD-vulnerable catecholaminergic brain nuclei from TgNM mice. A**, **C** From left to right, unstained macroscopic view, NM-sensitive high-resolution T1-weighted magnetic resonance imaging, hematoxilin-eosin (H&E) and Masson-Fontana (M-F) staining of aged healthy human postmortem brains (**A**), wt (**B**), and tgNM (**C**) mice. One micrograph from more than three high-magnification images from two representative specimens was selected and displayed. **C** (right panel), macroscopic view of a clarified tgNM brain. **B** Unstained macroscopic view (left) and NM-sensitive high-resolution T1-weighted magnetic resonance imaging (right) from wild-type (wt) mice acquired ex vivo without fat saturation. Unstained NM, brown (see also Movie S1). Dashed lines outline the SN/VTA and LC regions in (**A**–**C**). Scale bars: 20 μm (H&E), 5 μm (M-F). Macroscopic human midbrain image adapted from[19].

incipient neurodegenerative process in these animals. All these changes were accompanied by reductions of dopaminergic markers in striatal fibers (i.e., TH, dopamine transporter [DAT] and vesicular monoamine transporter 2 [VMAT2]), as measured by Western blot in dissected tissue and/or by optical densitometry in immunostained histological sections (Fig. S4C-D). While total dopamine (DA) levels were not changed in tgNM, as analyzed by ultra-performance liquid chromatography (UPLC) in striatal and SN-VTA tissue homogenates (Fig. S5A), these animals exhibited alterations in DA metabolic pathways, including decreased DA synthesis and increased catechol oxidation, the latter producing DA oxidized species acting as NM precursors[20] (Fig. S5B-C). In addition, striatal DA release, as assessed by in vivo microdialysis, was impaired in adult tgNM mice (Fig. S5D), an age concurring with the appearance of behavioral alterations in these animals.

Neuropathologically, NM-containing SNpc and VTA neurons from tgNM mice exhibited intracellular inclusion bodies as typically seen in PD and aged human brains, including cytoplasmic Lewy body (LB)-like inclusions and nuclear Marinesco bodies (MB) (Fig. 4A, B). Similar to humans[29], both LB and MB were immunopositive for p62 and ubiquitin, common components of neuropathological inclusions (Fig. 4A, B, Fig. S4E). More than 60% of p62-positive cytoplasmic inclusions were also immunopositive for alpha-synuclein (Fig. 4C). In contrast, as reported in humans[29], all MB were immunonegative for alpha-synuclein (Fig. 4B). Inclusion body formation was restricted to melanized neurons and was not observed in wt mice. The number of both

LB-like inclusions and MB was less prominent in old tgNM, coinciding with the intensification of neurodegenerative changes (i.e., eNM), thus suggesting that inclusion-containing neurons may be those that preferentially start to degenerate (Fig. 4A, B). In agreement with this, the number of neuronal inclusions in PD brains at advanced stages of the disease is much lower than that observed in early PD cases[30]. Inclusion formation in tgNM mice coincided with decreased expression of autophagy markers (Fig. S6A), both reflecting impaired proteostasis in NM-laden brain regions[19]. TgNM mice also exhibited early inflammatory changes in the SNpc and VTA, including increased numbers of GFAP-positive astrocytic cells and reactive Iba-1-positive microglial cells (Figs. 4D and S6B), the latter surrounding eNM as it occurs in postmortem PD brains[27,28]. Overall, progressive NM production in SNpc and VTA from tgNM mice is associated to age-dependent dopaminergic functional alterations and neuropathological changes linked to incipient neurodegeneration, as similarly observed in aged human brains and prodromal/early PD stages.

## Early LC pathology and noradrenergic neurodegeneration in tgNM mice

The LC is a highly melanized region in the pons that constitutes the main source of noradrenaline (NA) modulation in the brain. In addition, the LC is another major area consistently affected in PD, among the earliest sites of Lewy pathology and its degeneration is postulated to precede SNpc involvement and to account for non-motor PD symptoms, including sleep disturbances, anxiety-depression and

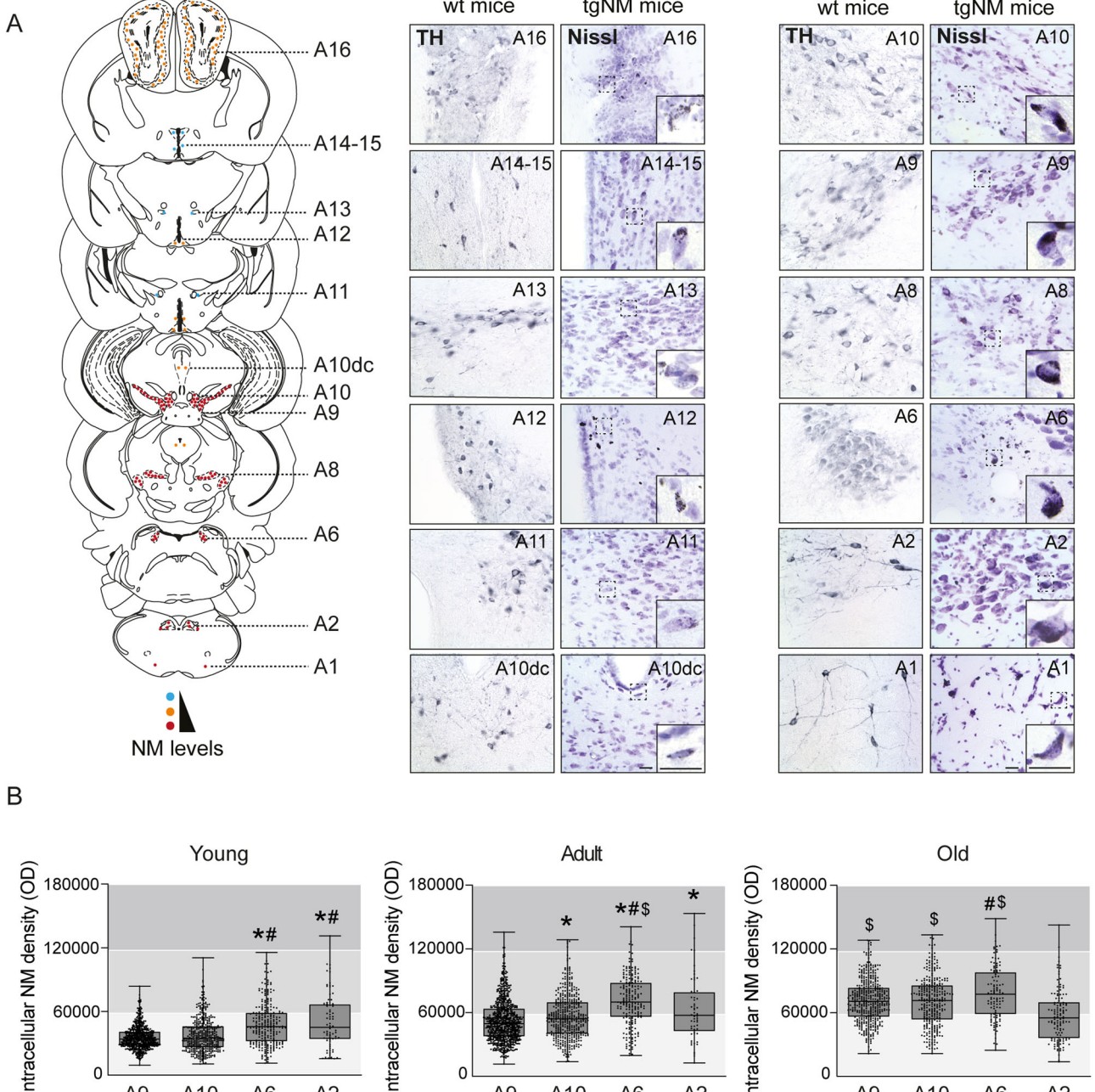

**Fig. 2 | Brain-wide NM distribution and accumulation of human-like NM in PD-vulnerable catecholaminergic brain nuclei from TgNM mice. A** Qualitative mouse brain atlas showing NM accumulation in all catecholaminergic groups in tgNM mice (see also Supplementary Table 1). Left, Mouse brain atlas. Number of dots represents the quantity of NM-accumulating cells and dot color represents the levels of intracellular NM from very low levels (blue) to higher levels (red). Image credit: Allen Institute. Right chatecholaminergic areas immunostained for TH in wt mice and Nissl-stained in tgNM mice. Unstained NM, brown. Scale bars: 25 μm, 25 μm (inset). **B** Quantification of intracellular NM levels in unstained SN/A9, VTA/A10, LC/A6, and DVC/A2 brain sections from tgNM mice *$p \le 0.05$ compared with A9, #$p \le 0.05$ compared with A10, $$p \le 0.05$ compared with A2. Box plots: median, first and third quartile, min-max values and individual dots for each neuron. Raw data, genotypes, ages, sample sizes and statistical analyses are provided as a Source Data file.

cognitive decline[7,30–32]. However, the LC is usually neglected in PD research. We next examined the consequences of NM buildup in nor-adrenergic LC neurons (Fig. S7A). First, we assessed the functional integrity of the LC in anxiety-related behavior (open field test), emotional memory (step-down test) and sleep (polysomnography) paradigms. Adult tgNM mice spent more time and traveled more distance in the periphery of an open field than their wt littermates (Fig. 5A), despite moving at a similar speed (Fig. S7B), indicative of an anxiety-related behavior. Old tgNM animals spent less time on a platform before stepping down and receiving an electric shock, indicative of a

deficit in the amygdala-dependent learning and memory process (Fig. 5B). Adult tgNM mice exhibited reduced amounts of both para-doxical sleep (PS) and slow-wave sleep (SWS) at the expense of wakefulness (Fig. S7C), indicative of an altered sleep-wake cycle. An in-depth analysis of PS evidenced significantly reduced numbers of bouts concomitant to a significant increase of their duration in both adult and old tgNM mice (Fig. 5C), indicative of an irreversible dysregulation of PS ultradian rhythm.

Stereological cell counts revealed a marked reduction of TH-positive cells in the LC of tgNM mice compared with wt littermates,

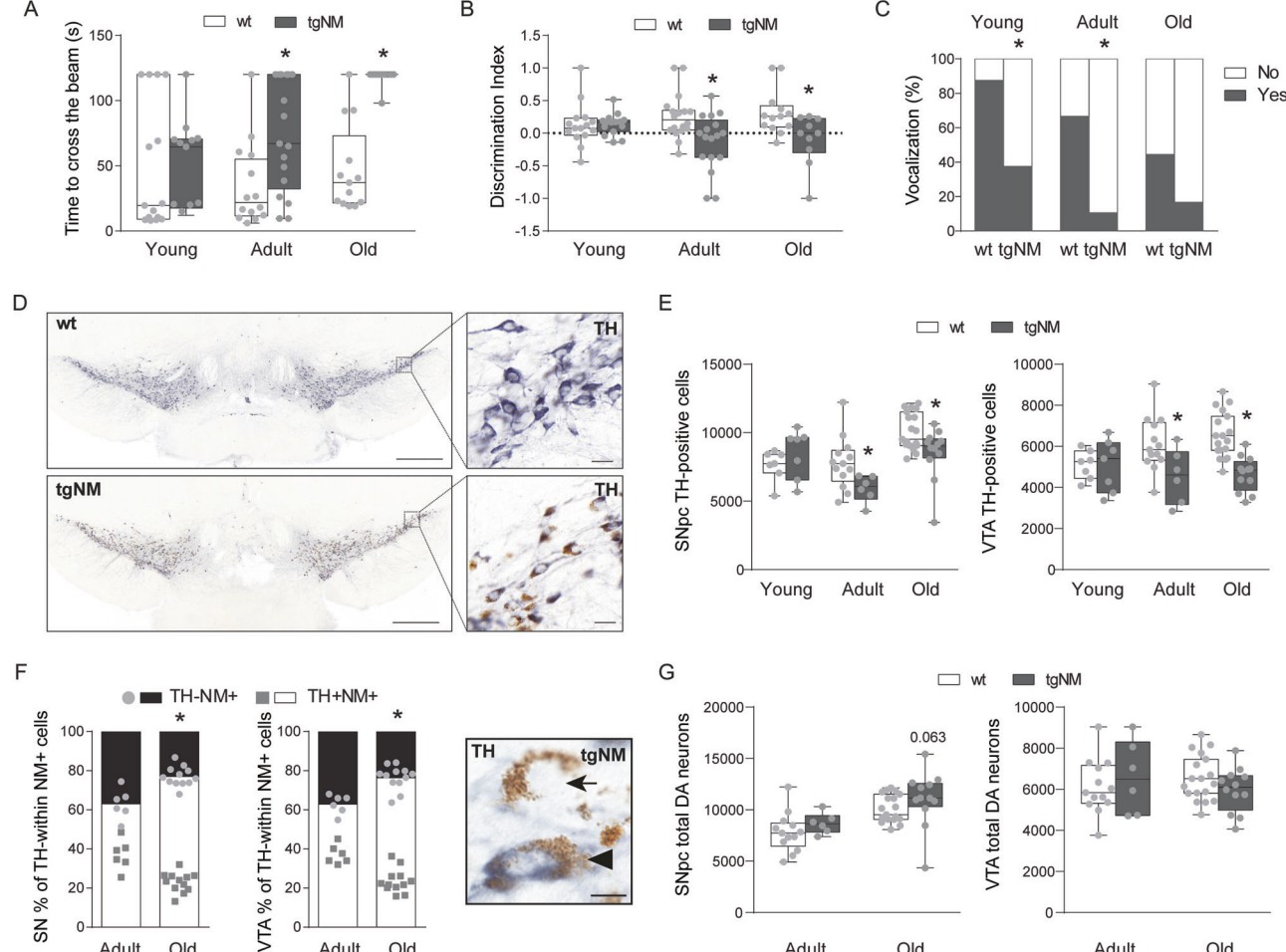

**Fig. 3 | Dopaminergic dysfunction in tgNM mice. A** Time to cross the beam in seconds (s). **B** Olfactory discrimination index. **C** Percentage of animals vocalizing. **D** SNpc and VTA immunostained sections. TH, blue; unstained NM, brown. Scale bars: 500 μm, 20 μm (inset). **E** Cell counts of SNpc and VTA TH-positive neurons. **F** Left, quantification of TH downregulation measured as the percentage (%) of TH-immunonegative neurons within the total population of NM-containing neurons in tgNM mice. Right, TH-immunostained SNpc showing TH⁻NM⁺ (arrow) and TH⁺NM⁺ (arrowhead) neurons. **G** Cell counts of total SNpc and VTA dopaminergic neurons. **A–C, E, F** *$p \leq 0.05$ compared with wt littermates. Box plots: median, first and third quartile, min-max values and individual dots for each neuron. Raw data, genotypes, ages, sample sizes and statistical analyses are provided as a Source Data file.

which was already present at a pubertal age (1 m) (Fig. 5D). In contrast to the SNpc and VTA above, the reduction in TH-positive cells in the LC corresponded to an actual neuronal loss as indicated by stereological cell counts of total noradrenergic NM-containing LC neurons (Fig. 5D). We also found abundant eNM at all ages concomitant with cell loss (Fig. 5E). LC neurodegeneration resulted in a significant reduction in total NA levels, already evident at a young age, in both LC and its projecting areas such as the prefrontal cortex (Fig. 5F). These animals also exhibited reduced NA synthesis and increased NA degradation (Fig. S7D), indicating altered NA neurotransmission. Melanized LC neurons also exhibited p62-positive cytoplasmic inclusions but almost no MB were detected (Figs. 5G and S7E). The highest number of LC cytoplasmic inclusions were found at pubertal ages and decreased afterwards (Fig. 5G), possibly due to the high degree of neuronal loss in this area with only a few remaining neurons left (Fig. 5D). Most of these inclusions were also immunoreactive for alpha-synuclein (Fig. S7F). Concomitant to neurodegeneration, the LC from tgNM mice exhibited marked inflammatory changes from a young age, including astrocytosis and microglial activation (Figs. 5H and S7G). Altogether, tgNM mice exhibit an early and extensive degeneration of the LC, linked to non-motor alterations, that precedes nigral dopaminergic dysfunction, similar to prodromal/early PD stages[7,31,32].

## Cholinergic and serotonergic alterations in tgNM mice

In addition to catecholaminergic pigmented areas, non-pigmented neurons also degenerate in PD, in particular cholinergic neurons from the nucleus basalis of Meynert (NBM) and the pedunculopontine nucleus (PPN)[33]. These two cholinergic nuclei are vastly connected with the brainstem PD-vulnerable catecholaminergic areas[34–37] (Fig. S8A), and their degeneration in PD may be related to the degeneration of the interconnected melanized neurons from the SNpc, VTA and LC[38,39]. We detected a significant decrease in the total number of choline acetyltransferase (ChAT)-positive neurons in the NBM and PPN nuclei in old tgNM mice (Figs. 6A, B and S8B). However, total acetylcholine (Ach) levels were not decreased in the prefrontal cortex of old tgNM animals, and they were actually increased in the PPN itself (Fig. 6C).

Another neurotransmitter system often altered in PD is the serotonergic system. Indeed, anxiety and depression are common non-motor symptoms affecting 30–35% of PD patients[40–42]. Variable degrees of cell death have been reported in the raphe nuclei from PD postmortem brains[33]. We assessed the integrity of the dorsal raphe (DR) nucleus in tgNM mice. Stereological cell counts of tryptophan hydroxylase (TPH)-positive serotonergic DR neurons did not show statistically significant differences in old tgNM mice, although these numbers were overall lower than in wt littermates (Fig. 6D). Of note,

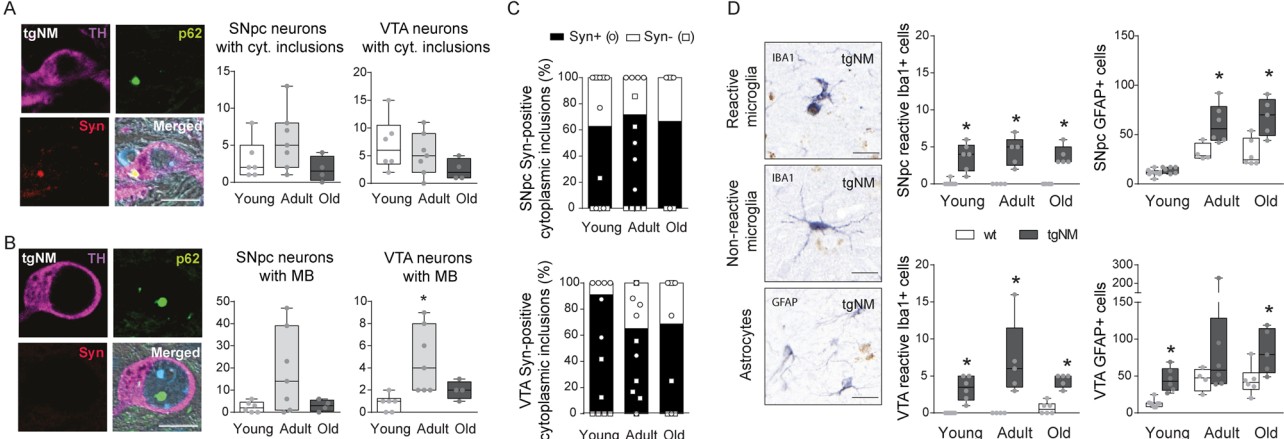

**Fig. 4 | PD-like neuropathology in tgNM mice. A, B** Left, SNpc and VTA sections exhibiting NM-laden neurons with cytoplasmic (**A**) and nuclear (**B**) inclusions. TH, purple; p62, green; alpha-synuclein (Syn), red; Hoechst (blue); NM, dark gray. Scale bar: 10 μm. Right, quantification of SNpc and VTA NM-laden neurons with p62-positive cytoplasmic (**A**) and nuclear (**B**) inclusions. **C** Percentage of SN-VTA cytoplasmic inclusions positive or negative for Syn in tgNM at different ages. **D** Left, reactive (top) and non-reactive (middle) Iba-1-positive microglia (blue) and GFAP-positive astrocytes (bottom, blue). Scale bar: 20 μm. Right, quantification of reactive Iba-1- and GFAP-positive cells in SNpc and VTA sections. **B** *$p \leq 0.05$ compared with young tgNM. **D** *$p \leq 0.05$ compared with wt littermates. Box plots: median, first and third quartile, min-max values and individual dots for each neuron. Raw data, genotypes, ages, sample sizes and statistical analyses are provided as a Source Data file.

some TPH-positive DR cells (~3%) contained NM, which is compatible with these neurons being also catecholaminergic, as previously described[43,44]. No significant changes were found in serotonin (5-HT) levels in the DR nor in the PFC of tgNM mice (Fig. 6E). Still, tgNM mice exhibited increased immobilization time in the tail suspension test, indicative of a depressive-like behavior[45] (Fig. 6F).

### NM accumulation in medullary nuclei and autonomic dysfunction in tgNM

Pigmented catecholaminergic neuronal groups in the caudal medulla have also been reported to degenerate in PD, in particular NM-containing neurons of the dorsal motor nucleus of the vagus region[46–49]. These neurons, which are part of the dorsomedial medullary A2 group and present both noradrenergic and adrenergic cells, intermingle with cholinergic neurons of the dorsal motor nucleus of the vagus nerve (DNV) and nucleus of the solitary tract (NTS) to constitute an overlapped structure referred to as the DVC, which altogether provide the major integrative center for the mammalian autonomic nervous system. We next characterized the viability of DVC pigmented neurons in tgNM mice (Figs. 7A and S9A). Stereological cell counts of TH-positive and total catecholaminergic neurons in DVC sections showed no consistent decreases in tgNM mice compared with wt littermates (Fig. S9B). However, only a small percentage (~10%) of TH-positive neurons in the DVC were actually pigmented. Hence, we evaluated only the pigmented TH-positive neurons and found a significant decrease of these cells with age (Fig. 7B). In adult tgNM mice, this decrease mostly corresponded to a phenotypic TH down-regulation, as the number of total pigmented cells was not yet reduced at this stage (Fig. 7C). Reduced number of pigmented TH-positive DVC neurons was accompanied by extensive eNM and microglial activation (Figs. 7D, E and S9C). A few cytoplasmic inclusions were also detected in melanized DVC neurons, but only at pubertal ages preceding phenotypic loss in adult ages (Fig. 7F). In contrast, ChAT-positive DVC cells were not affected in tgNM mice (Fig. S9D).

To evaluate the functional consequences of medullary DVC alterations, we assessed autonomic parameters (cardiovascular, respiratory and gastrointestinal) in tgNM mice (Fig. 8A). Old tgNM mice showed decreased heart rate (Fig. 8B), despite no apparent alterations in blood pressure (Fig. 8C). These animals also exhibited increased respiratory rate by measuring diaphragmatic movements

(Fig. 8D). In contrast, no changes in gastrointestinal function were found in tgNM mice by measuring intestinal transit time with oral red carmine administration (Fig. 8K). Concomitant with these signs of peripheral autonomic dysfunction, we detected a significant decrease in DA and NA levels, as well as altered synthesis and degradation, in the DVC of tgNM mice (Figs. 8F and S9E). In addition, alterations in NA and Ach levels were also detected in vagal-innervated peripheral organs of tgNM mice like the heart and intestines (Fig. 8F). Dysautonomic disorders, in particular cardiovascular and respiratory, are reported to be more frequent causes of death in PD than in age-matched controls[50,51]. We evaluated life expectancy in tgNM mice and detected a decreased lifespan compared to their wt counterparts (median survival of TgNM mice: 20.7 m; Fig. 8G). Overall, in addition to the brain-related alterations reported above, our results reveal that tgNM mice also recapitulate some of the peripheral autonomic alterations occurring in PD.

## Discussion

Modelling the physiopathology of neurodegenerative diseases remains a difficult task. Several animal models have been developed to understand disease pathogenesis and test new drug candidates. However, neurodegenerative diseases are highly heterogeneous diseases involving several factors and pathways. Therefore, none of the existing models replicates the entire spectrum of clinical and neuropathological features occurring in neurodegenerative diseases. Here we provide an animal model that fills a significant gap in the field by incorporating in rodents a human factor intimately linked to brain aging and neurodegeneration such as NM, which has been so far neglected in experimental preclinical in vivo research. Indeed, because NM is not spontaneously produced in the most commonly used laboratory animals, such as rodents, work on this pigment has been so far limited to a small number of studies using scarcely available human brain tissue. Consequently, many aspects of the biology of NM remain to be elucidated. While NM shares some features with melanin found in the periphery (e.g., skin, hair), peripheral melanin is structurally, functionally and compositionally different from NM. Peripheral melanogenesis occurs within specialized cells (i.e., melanocytes) through a biosynthetic pathway enzymatically driven by TYR as the key, rate-limiting enzyme[52]. In contrast, the presence of TYR in the human brain, that we and others have reported to be expressed at very low levels[19,53–56], is still disputed[55,57–60]. In fact, it is widely assumed that NM

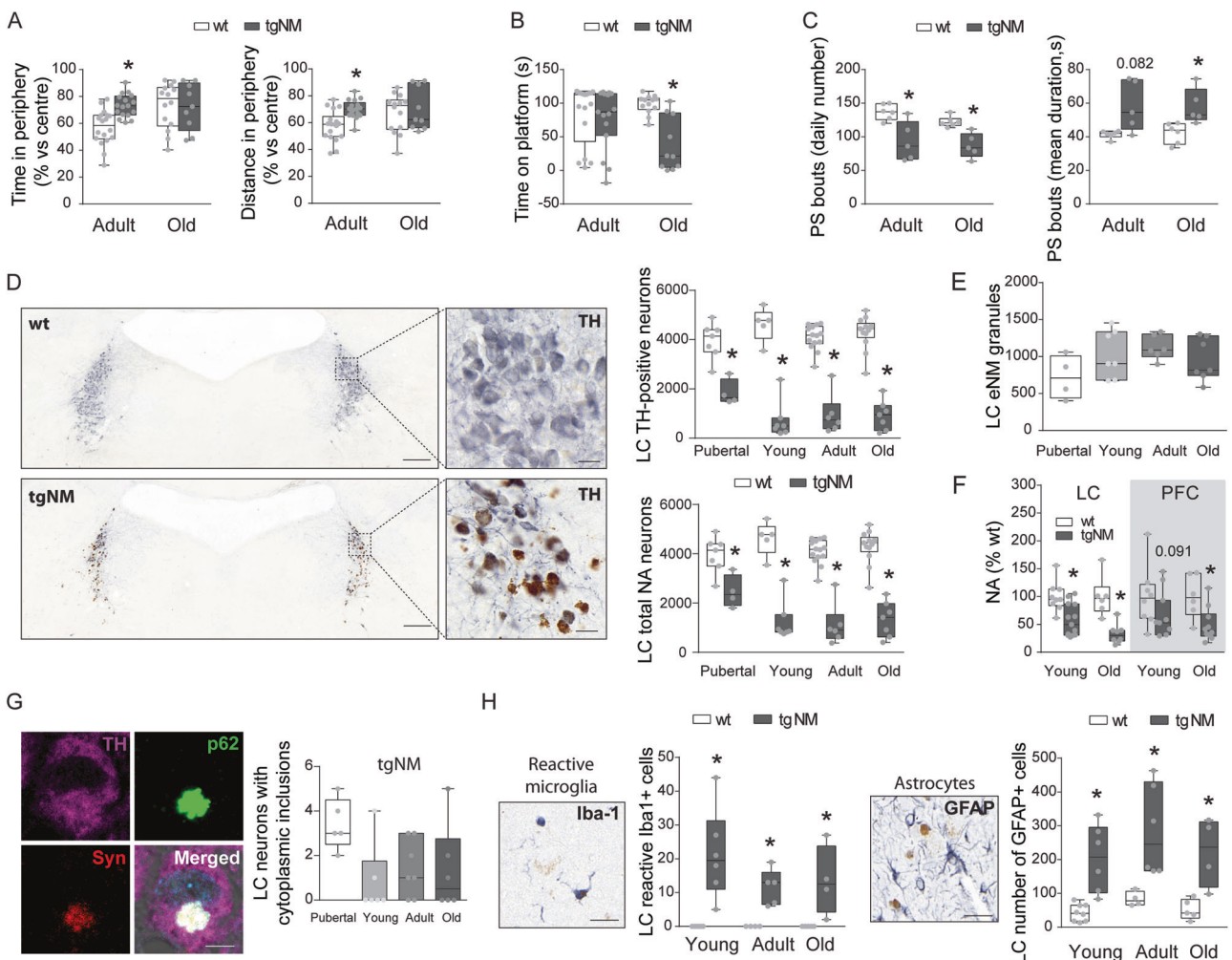

**Fig. 5 | Noradrenergic neurodegeneration and PD-like neuropathology in tgNM mice. A** Percentage of time spent and distance traveled in the periphery compared with the center of an open field. **B** Time spent on the platform of a step-down test. **C** Number and mean duration (s, seconds) of paradoxical sleep (PS) bouts. **D** Left, LC sections immunostained for TH. TH, blue; unstained NM, brown. Scale bars: 200 μm, 20 μm (inset). Right, cell counts of LC TH-positive neurons and total NA neurons at different ages. **E** Number of eNM granules in tgNM mice. **F** LC and prefrontal cortex (PFC) NA levels in young/old tgNM and wt mice. **G** Left, LC section exhibiting a NM-laden neuron with a cytoplasmic inclusion. TH, purple; p62, green; alpha-synuclein (Syn), red; Hoechst (blue); NM, dark gray. Scale bar: 5 μm. Right, quantification of LC NM-laden neurons with p62-positive cytoplasmic inclusions in tgNM mice. **H** Images and quantification of reactive Iba-1- and GFAP-positive cells in LC sections. **A−D**, **F**, **H** *$p \le 0.05$ compared with wt littermates. Box plots: median, first and third quartile, min-max values and individual dots for each neuron. Raw data, genotypes, ages, sample sizes and statistical analyses are provided as a Source Data file.

may be produced by spontaneous nonenzymatic dopamine auto-oxidation[61]. However, experimentally increasing dopamine or oxidized dopamine levels in rodents, either with chronic L-dopa treatment[62,63] or by genetically enhancing TH activity[64], is not sufficient by itself to produce NM in these animals, as it might be expected if NM represents a mere process of DA autoxidation. Remarkably, synthetic melanin produced in a test tube by oxidizing dopamine with TYR resembles more closely NM, in terms of absorbance spectra, elemental composition and melanic components, than synthetic melanin obtained by dopamine autoxidation[65,66]. We have previously corroborated, and extended it here, that the neuronal pigment induced by TYR over-expression in rodent catecholaminergic neurons is indeed very similar to human NM, including: (i) its detection by NM-sensitive magnetic resonance imaging, which reflects NM's avid binding of iron[19]; (ii) its chemical composition by UPLC-MS/MS, comprising all NM's melanic components[20,67]; or (iii) its ultrastructure by electron microscopy, exhibiting an electron-dense matrix associated to characteristic lipid droplets within autophagic compartments[19]. This complex structure, including melanic, lipid, peptide and inorganic components contained inside special autolysosomes is a defining hallmark of true NM as

opposed to peripheral melanin[68]. Therefore, independently of the current debate around the endogenous expression and role of TYR in the human brain, the neuronal pigment produced in tgNM mice can be considered as analogous to human NM, thereby supporting the relevance of our approach to study the impact of human-like NM pigment on age-dependent neuronal function and viability in vivo.

The functional significance of NM production, if any, is not currently known. This pigment was long considered just a waste product of catecholamine metabolism devoid of any physiological function. More recently, it has been hypothesized that NM could play a neuro-protective role by removing excessive cytosolic catecholamine oxidized species, chelating potentially toxic metals or sequestering environmental neurotoxins[69]. However, a putative protective effect of NM has never been formally demonstrated, probably because of the lack of proper experimental models, and thus remains speculative to this day. In any case, even if we consider NM synthesis to be an initially beneficial process, this is compatible with a deleterious effect of its long-term age-dependent accumulation, by ultimately interfering with normal cell function and proteostasis in addition to an excess production of potentially toxic oxidized catechol intermediates, as we

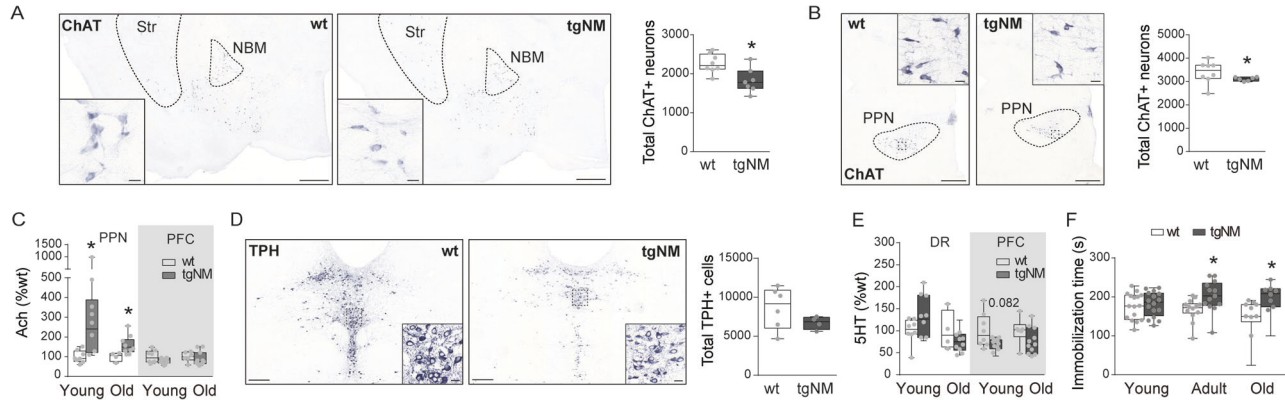

**Fig. 6 | Cholinergic and serotonergic alterations in tgNM mice. A**, **B** Left, NBM and PPN sections immunostained for choline acetyltransferase (ChAT). ChAT, blue; unstained NM, brown. Scale bars: 500 μm, 20 μm (inset). Right, cell counts of NBM and PPN ChAT-positive neurons in old tgNM and wt mice. **C** PPN and prefrontal cortex (PFC) acetylcholine (Ach) levels. **D** Left, dorsal raphe (DR) sections immunostained for tryptophan hydroxylase (TPH). TPH, blue; unstained NM, brown.

Scale bars: 200 μm, 20 μm (inset). Right, Cell counts of DR TPH-positive neurons in old tgNM and wt mice. **E** DR and PFC serotonin (5HT) levels. **F** Immobilization time in the tail suspension test (s, seconds). **A**–**C**, **F** *$p \le 0.05$ compared with wt littermates. Box plots: median, first and third quartile, min-max values and individual dots for each neuron. Raw data, genotypes, ages, sample sizes and statistical analyses are provided as a Source Data file.

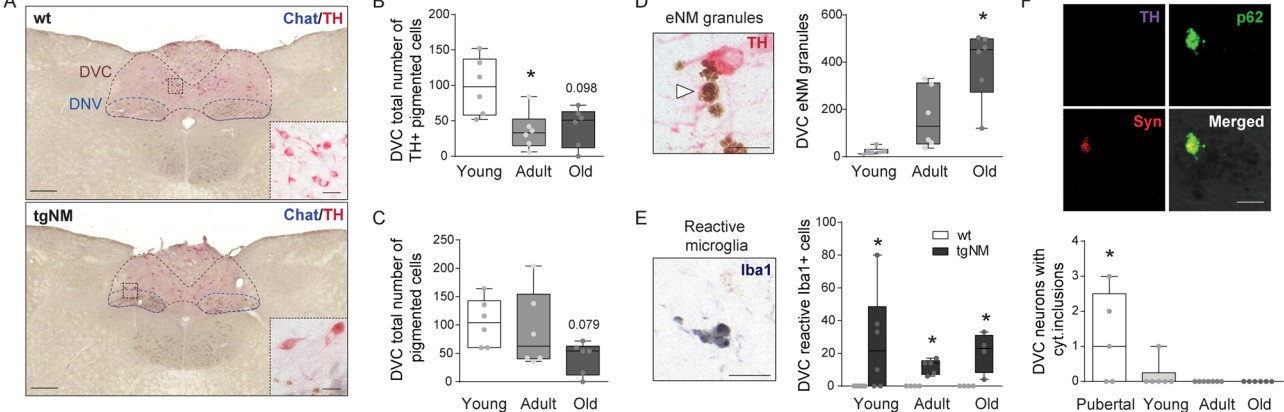

**Fig. 7 | Medullary catecholaminergic nuclei alterations in tgNM mice.**
**A** Medullary sections immunostained for TH and ChAT. TH, red; ChAT, blue; unstained NM, brown. Scale bars: 200 μm, 10 μm (insets). Dorsal motor nucleus of the vagus nerve, DNV; dorsal vagal complex, DVC. **B**, **C** Cell counts of TH-positive (**B**) and total (**C**) pigmented catecholaminergic cells. **D** Left, image of eNM granules (white arrowhead). Scale bar: 25 μm. Right, number of eNM granules in A2 neuronal group in tgNM mice. **E** Images and quantification of Iba-1-positive reactive microglia in tgNM and wt mice. **F** *Top*, DVC section exhibiting a NM-laden neuron with a

cytoplasmic inclusion. TH, purple; p62, green; alpha-synuclein (Syn), red; Hoechst (blue); NM, dark gray. Scale bar: 5 μm. *Bottom*, quantification of DVC NM-laden neurons with p62-positive cytoplasmic inclusions in tgNM mice. **B**, **D** *$p \le 0.05$ compared with young tgNM. **E** *$p \le 0.05$ compared with wt littermates. **F** $p \le 0.05$ compared with adult and old tgNM. Box plots: median, first and third quartile, min-max values and individual dots for each neuron. Raw data, genotypes, ages, sample sizes and statistical analyses are provided as a Source Data file.

observed in NM-producing rodents[19,20,70]. Supporting this concept, reduction of intracellular NM accumulation in AAV-TYR-injected rats, either by boosting NM cytosolic clearance with autophagy activator TFEB or by delaying age-dependent NM production through VMAT2-mediated enhancement of dopamine vesicular encapsulation, resulted in a major attenuation of their neurodegenerative phenotype, both at the behavioral and neuropathological level[19,20]. In contrast, in absence of associated NM formation, TYR expression was not apparently toxic per se, as indicated by a lack of NM production and degeneration of TYR-expressing gamma-aminobutyric acid nigrotectal neurons in AAV-TYR-injected rats[19].

Our results in tgNM mice suggest that age-dependent brain-wide pigmentation, as it occurs in humans, may be sufficient to ultimately trigger a progressive neurodegenerative phenotype with motor and non-motor deficits, affecting multiple neuronal systems in the brain, beyond the SNpc, as well as peripheral organs, all reminiscent of what is observed in prodromal/early PD stages. Brain pigmentation in tgNM

mice follows a caudorostral gradient of accumulation that parallels the occurrence of neuropathological alterations observed in PD, reaching higher levels of pigmentation and neuronal pathology earlier in the medulla and pons than in midbrain nuclei[11,47,71]. In this context, the caudorostral gradient of alpha-synuclein pathology reported in PD brains[30] may be linked to a gradient of increasing intracellular NM levels promoting Lewy pathology. In agreement with this, alpha-synuclein has been reported to redistribute to the NM pigment in early stages of PD and become entrapped within NM granules, which may predispose melanized neurons to precipitate alpha-synuclein around pigment-associated lipids under oxidative conditions, such as those linked to NM formation[18,72,73]. Our findings also suggest that the neuroinflammatory changes found in human non-diseased aged and PD brains[8,25,74,75] may actually start prior to overt neurodegeneration and be linked to the presence of incipient eNM released from early degenerating neurons. Also, as we previously reported[19], NM intracellular accumulation results in a general proteostasis failure leading

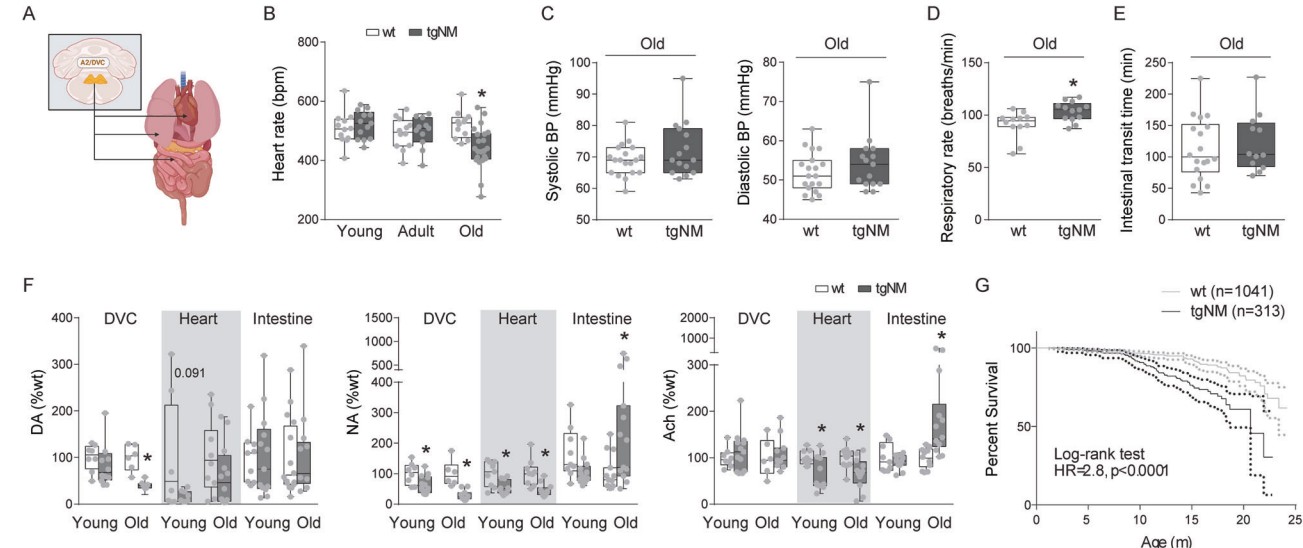

**Fig. 8 | Autonomic dysfunction and decreased lifespan in tgNM mice.**
**A** Schematic diagram of the peripheral A2 innervation sites. **B** Heart rate (bpm, beats per minute) in tgNM and wt mice. **C** Systolic and diastolic blood pressure in old tgNM and wt mice. **D** Respiratory rate (number of breaths/min) in old tgNM and wt mice. **E** Intestinal transit time (min, minutes) in old tgNM and wt mice. **F** Levels of DA, NA and Ach in DVC and peripheral organs from tgNM and wt mice. **G** Survival analysis of tgNM mice (black) compared with wt (gray). Dotted lines represent 95% confidence interval for each genotype. m, months. **B–G** *$p \le 0.05$ compared with wt littermates. Box plots: median, first and third quartile, min-max values and individual dots for each neuron. Raw data, genotypes, ages, sample sizes and statistical analyses are provided as a Source Data file.

to the formation of MB and LB-like inclusion bodies in different pigmented brain areas of tgNM mice. Supporting that inclusion-containing neurons are those that preferentially degenerate in PD, the largest numbers of inclusions were found prior to overt neurodegeneration. Finally, our data reveals noradrenergic neurodegeneration already at a pubertal age in tgNM mice, supporting that noradrenergic denervation might precede nigral degeneration as well in human PD. Symptoms associated with noradrenergic deficits due to LC neurodegeneration in humans include several non-motor alterations such as sleep disorders, anxiety, depression, cognitive alterations and autonomic dysfunction[76,77]. These are reproduced in tgNM mice due to the early LC degeneration concomitant with high levels of NM accumulation already at young ages. In line with this, significant noradrenergic denervation has indeed been observed in patients with idiopathic rapid eye movement (REM) sleep behavior disorder (RBD)[78,79], which is considered a prodromal form of PD.

Lastly, tgNM mice also showed alterations in non-catecholaminergic non-pigmented neuronal systems such as the cholinergic and serotoninergic (i.e., cell counts, neurotransmitter levels and/or behavioral readouts), which are also at risk in PD. Cholinergic degeneration in the PPN has been linked to gait and balance disturbances[80,81] and cholinergic cell loss in the NBM has been linked to cognitive impairment in PD[82]. While studies evaluating the serotoninergic Raphe nuclei show conflicting results[33], decreased PET functional imaging studies and decreased serotoninergic metabolites have been reported in PD and linked to depressive-like behaviors[83].

Because all humans accumulate NM with age, our results in NM-producing rodents imply that all humans could potentially develop NM-linked neurodegenerative changes if they were to live long enough to reach pathogenic levels of intracellular NM. However, and focusing on PD, only a fraction of the aging population actually develops overt neurodegenerative disease (e.g., up to ~5% of the population over 60 in the case of PD, increasing with age[84]. Yet, in absence of clinically diagnosed PD: (i) 10–30% of apparently healthy people older than 60 years already exhibit Lewy pathology in their melanized brainstem nuclei (i.e., incidental Lewy body disease, who are considered to represent early, presymptomatic stages of PD)[85]; (ii) parkinsonian signs

such as bradykinesia, stooped posture and gait disturbance are common in elderly individuals, reaching a prevalence of up to 50% in individuals over 80 years of age[86,87]; (iii) multiple reports have documented an age-related loss of pigmented nigral neurons, estimated at about 5–10% per decade, in otherwise healthy individuals[8,11,12]; (iv) brains from aged individuals commonly exhibit a downregulation of dopaminergic phenotypic markers within melanized nigral neurons, which reflects neuronal dysfunction at early stages of neurodegeneration, as well as abundant eNM released from dying neurons associated with sustained microglial activation, compared to young adult brains[8–10]. These observations, combined with our results in NM-producing rodents, suggest that progressive NM accumulation may not be just an innocuous consequence of normal catecholaminergic neuronal aging, as it is currently regarded, but underlie incipient age-dependent functional/degenerative changes in these neurons that would clinically manifest as overt neurodegenerative disease once a certain level of NM accumulation and subsequent cell dysfunction/death is reached. In agreement with this, intracellular NM levels are increased in dopaminergic nigral neurons of PD patients and subjects with incidental Lewy body disease, compared to age-matched healthy individuals, indicating that they may have reached earlier a pathogenic threshold of intracellular NM accumulation[19,55,73]. Additional elements, such as genetic or environmental factors, could potentially accelerate or trigger pathology in these already vulnerable NM-filled neurons. Age-dependent brain pigmentation may thus represent a key molecular substrate underlying the well-established association between aging and increased risk of neurodegenerative disease.

While some important questions remain to be solved, such as understanding the factors that influence brain pigmentation in humans, our results collectively point to age-dependent NM accumulation as a key pathogenic factor driving PD pathology. Overall, the tgNM mouse model reported here, now readily available to the entire scientific community, provides the opportunity to experimentally address all of these questions in a human-relevant context. These humanized animals recapitulate the biological and temporal complexity of age-dependent neurodegeneration, affecting numerous brain neurotransmission systems and tissues in the body leading to a

myriad of motor and non-motor symptoms. Because of the prodromal nature of these changes, these animals could be combined with additional disease-relevant genetic or environmental manipulations. In addition, these animals could also be used to test potential therapeutic strategies in a human disease state-informed context at different stages of the disease, which should facilitate the translation of findings to humans. Furthermore, the possibility to use a non-invasive technique like MRI to monitor NM accumulation in vivo represents a useful tool for further in vivo studies testing therapeutic approaches in these animals. The failure, to date, of many rodent model-based findings to translate to clinical utility in the context of age-related neurodegenerative disease strengthens the importance of introducing into in vivo research a human factor so intimately linked to brain aging and neurodegeneration such as NM.

## Methods

This study complies with all relevant ethical regulations. All procedures involving human samples were conducted in accordance with guidelines established by the BPC (CPMP/ICH/135/95) and the Spanish regulation (223/2004) and approved by the Vall d'Hebron Research Institute (VHIR) Ethical Clinical Investigation Committee (PR(AG)370/2014). All the experimental and surgical procedures using animals were conducted in strict accordance with the European (Directive 2010/63/UE) and Spanish laws and regulations (Real Decreto 53/2013; Generalitat de Catalunya Decret 214/97) on the protection of animals used for experimental and other scientific purposes. Protocols were approved by the Vall d'Hebron Research Institute (VHIR) Ethical Experimentation Committee and the Generalitat de Catalunya (Protocol 11442), as well as the local CELYNE Ethics Research Committee of the Université Claude Bernard Lyon 1 (Protocol APAFIS#20701) for the sleep study.

### Human postmortem brain tissue

For NM characterization in the human brain paraffin-embedded midbrain sections from healthy control subjects ($n = 4$, mean age = 80.75 ± 1.65 years) were provided by the Neurological Tissue BioBank at *IDIBAPS-Hospital Clinic* (Barcelona) and the *Biobanco en Red de la Región de Murcia* (BIOBANC-MUR). Standard hematoxylin-eosin and Masson-Fontana staining (see below) were performed on 5 μm-thick paraffin-embedded SNpc and LC sections. NM-MRI images were provided by Dr. Alex Rovira (Neuroradiology section, Vall d'Hebron Hospital, Barcelona, Spain). Autopsy representative images from midbrain and pons (control subjects aged 62 and 51 years, respectively) were provided by Dr. Ellen Gelpi (Division of Neuropathology and Neurochemistry, Department of Neurology, Medical University of Vienna, Austria).

### TgNM mouse colony

Transgenic mice B6.Cg-Tg(Th-TYR)26Mvila/Mvila (RRID:MGI:7730751) were obtained by pronuclear microinjection of the human tyrosinase complementary DNA (cDNA) fused to the rat tyrosine hydroxylase promoter (Fig. S10) into C57Bl6-SJL (RRID:MGI:5656718) mouse zygotes (Center for Animal Biotechnology and gene Therapy [CBA-TEG]). The plasmid containing the cDNA for TYR was kindly provided by Dr. Takafumi Hasegawa (Department of Neurology, Tohoku University School of medicine, Sendai, Japan)[88]. Animals exhibited a normal reproductive performance (litter size ± SD = 6.2 ± 4.5). Adult/old TgNM mice exhibited an increased body weight of 21% (average) compared to wt littermates. Mice were backcrossed for 8–10 generations using C57BL/6 J mice (Charles River; RRID:MGI:3028467) and maintained in heterozygosis. Animals were housed two to five per cage with *ad libitum* access to food and water during a 12 hours (h) light/dark cycle (light-on 8 a.m.). To reduce the number of animals used for the study, age variable was prioritized over sex variable, the latter not being included in the experimental design. Male and female mice were evenly assigned to all experimental groups to avoid sex bias. No post hoc sex analysis was performed because of low sample size.

### Genotyping

DNA was routinely extracted from mouse ear punches with the kit AccuStart™ II Mouse Genotyping Kit (QuantaBio, #95135) and a PCR was performed by mixing 1 μl of DNA with the primers TYR-Forward (TTCAGACCCAGACTCTTTTCAA), TYR-Reverse (GCTGCTTTCTCTTGTGACGA) at a concentration of 500 nM using a 9800 Fast Thermal Cycler (Applied Biosystems). Amplification product was loaded into 1.8% agarose gel and visualized using a GelDoc XR (BioRad). Male and female wt and tgNM mice were randomly distributed into the different experimental groups and wt/tgNM animals were processed at once to minimize bias.

### Survival analysis and reproduction

All animals were followed throughout their lifespan and a survival analysis (log-rank test and Gehan-Breslow-Wilcoxon test) was performed considering all-natural deaths as positive events and all animals used for experimental purposes as censored data. Animals used for experimental purposes were chosen randomly and for reasons unrelated to their health status, thus avoiding a potential bias due to censoring[89]. For reproduction assessment, all heterozygous matings ($n = 147$) were registered with a mean of 6.2 ± 4.5 pups per litter (mean ± SD).

### Magnetic resonance imaging

Proton nuclear magnetic resonance imaging (¹H-MRI) studies were performed at the joint nuclear magnetic resonance facility of the *Universitat Autònoma de Barcelona* and *Centro de Investigación Biomédica en Red-Bioingeniería, Biomateriales y Nanomedicina* (CIBER-BBN) (Cerdanyola del Vallès, Spain), Unit 25 of NANBIOSIS. Experiments were conducted on a 7 T Bruker BioSpec 70/30USR scanner (Bruker BioSpin GmbH, Ettlingen, Germany) equipped with a mini-imaging gradient set (400mT/m) and using a 72-mm inner diameter linear volume coil as a transmitter and a dedicated mouse brain surface coil as a receiver, which provides a fairly homogenous signal intensity over the brain volume. All MRI data were acquired and processed on a Linux computer using Paravision 5.1 software ((RRID:SCR_001964; http://www.bruker.com/service/support-upgrades/software-downloads/mri.html; Bruker BioSpin GmbH, Karlsruhe, Germany). Mice ($n = 2$ wt and $n = 2$ tgNM) were anesthetized (1.5–2% isoflurane in 1 L/min oxygen for maintenance) and placed into an animal bed with bite-bar and ear-bars for optimal head immobilization. An animal monitoring and control system (SA Instruments, Stony Brook, NT) was used to control the respiration rate (50–100 bpm) and core body temperature (37 ± 1 °C) were maintained via integrated water hoses in animal's bed during the whole session. Low resolution T2-weighted fast spin-echo images were initially obtained in axial, sagittal and coronal planes to be used as reference scout images. Imaging parameters for these images are: effective echo time (TEeff) = 36 ms; repetition time (TR) = 2 s; echo train length (ETL) = 8; field of view (FOV) = 1.92 × 1.92 cm²; matrix size (MTX) = 128 × 128; slice thickness (ST) = 1 mm; gap between slices (gap)= 0.1 mm; number of slices (NS) = 15 -axial, 9 -sagittal, 5 -coronal; number of averages (NA) = 1. T2-weighted fast spin-echo axial images covering the whole brain were acquired with TEeff=36 ms; TR = 4.2 s; ETL = 8; FOV = 1.92 × 1.92 cm²; MTX = 256 × 256; ST = 0.5 mm; gap = 0.05 mm; NS = 23; NA = 4; Acquisition time (Acqt)=6 min 43 s. High-resolution T1-weighted spin-echo images were acquired afterwards in the axial plane with and without fat saturation pulses to see if we had better NM contrast in the brain by suppressing the subcutaneous fat, which might saturate the image because it is very intense in the T1w images and could be obscuring NM contrast. For fat suppression, a frequency-selective 90 degree gauss512 pulse was globally applied to a frequency offset of −3.5 ppm

relative to water (fat suppression bandwidth=1051.16 Hz). Transverse magnetization produced by the pulse was suppressed with a gradient spoiler. Two slice packages of 7 contiguous slices each were acquired containing the SNpc and LC regions and using the following parameter: TE = 7 ms; TR = 500 ms; NA = 56; FOV = 1.92 × 1.92 cm$^2$; MTX = 128 × 128; ST = 0.25 mm; thus, resulting in a spatial resolution of 150 × 150 × 250 µm$^3$; Acqt = 1 h per image. For T1w images, the effective Spectral Bandwidth used was 73529.4 Hz. After euthanasia, at 16 months of age, the brains were dissected and embedded in a 1% agarose block to perform ex vivo imaging using the same adjusted parameters. The ex vivo imaging results confirmed the location of hyperintense areas observed in vivo to the SN and LC brain regions. The images shown in Fig. 1 correspond to images acquired ex vivo without fat saturation.

## Brain processing

Animals were deeply anesthetized with sodium pentobarbital intraperitoneally and then perfused through the left ventricle with saline (0.9% [wt/vol]) at room temperature (RT), followed by ice-cold formaldehyde solution 4% phosphate buffered for histology (Panreac). Brains were removed and post-fixed for 24 h in the same fixative and subsequently processed for paraffin embedding following standard procedures or cryoprotected for 24–48 h in 30% sucrose at 4 °C and frozen. Sectioning was performed with a sliding microtome (Leica, Germany) at 5 µm-thickness for paraffin samples or in a cryostat at 30 µm-thickness for frozen samples (Leica, Germany).

## Brain clarification

The protocol used was modified from[90]. Briefly, brains were embedded in 1% low melting point agarose and dehydrated in methanol. Once dehydrated the brains were incubated during 4 hours shaking in 66% dichloromethane (DCM)/33% methanol at room temperature. Then the brains were incubated with shaking in 100% DCM 30 minutes twice to complete the delipidation process. Finally, the brains were chemically cleared with BABB (a combination of 1 part of benzyl alcohol and 2 parts of benzyl benzoate). The brains were subsequently immersed in a BABB-filled chamber for Optical projection tomography (OPT) imaging. The brains were scanned with transmission light with three different filters (a cyan fluorescent protein -CFP-filter: emission 460–500 nm, a green fluorescent protein -GFP- filter: emission 500–550 nm and a DSR filter: emission >610 nm) using a home-built OPT imaging system mounted on a Leica MZ 16 FA microscope[91]. The brain images were visualized using the open source software Fiji[92] (RRID:SCR_002285; http://fiji.sc/).

## In situ hybridization

Cryosections were mounted on slides and fixed for 10 min in 4% paraformaldehyde, washed in PBS and permeabilized with Proteinase K for 5 min (1 µg/mL Proteinase K, 50 mM Tris·HCl, and 5 mM EDTA). The sections were then refixed in 4% paraformaldehyde for another 10 min, followed by rinsing in PBS and acetylation for 10 min with 590 mL of H2O, 8 mL of triethanolamine, 1 mL of 37% HCl, and 1.5 mL of acetic anhydride. After three more rinses in PBS, the slides were incubated for 3 h at room temperature for prehybridization (50% formamide, 5× SSC, 5× Denhardt's solution, 250 µg/mL Baker's yeast RNA, and 500 µg/mL salmon sperm DNA). Hybridization was performed overnight at 72 °C in 120 µL prehybridization mix with 2 µL of digoxigenin-labeled sense (control probe not complementary) and antisense (test probe complementary) probes for tyrosinase. Riboprobes were generated by PCR and labeled with the DIG RNA labeling kit from Roche (11175033910). The next day, slides were washed in 72 °C heated solutions of 5× SSC and 0.2× SSC, and then equilibrated in B1 buffer [0.1 M Tris·HCl (pH 7.5) and 0.15 M NaCl]. After preincubation for 1 h in 10% heat-in-activated FCS in B1, the sections were incubated overnight with antidigoxigenin-AP Fab

fragments (Roche, diluted 1:5000 in B1 with 1% heat-inactivated FCS). After washing in B1, the sections were placed in a solution of 0.1 M Tris·HCl (pH 9.5), 0.1 M NaCl, and 50 mM MgCl2 (B2 buffer). Finally, mRNA localization was visualized after overnight incubation with 0.2 mM NBT/BCIP (Roche) and 0.24 µg/mL levamisole in 10% PVA/B2 solution protected from light. The color reaction was terminated by water and mounted in Aquatex (Merck). Sections were analyzed and photographed with a Zeiss Imager.D1 microscope coupled to an AxioCam MRc camera and with Pannoramic 250 Flash III (3D Histech), and processed with ZEN 2011 software (Zeiss, Germany; RRID:SCR_013672; https://www.zeiss.com/microscopy/en/products/software/zeiss-zen.html) and Caseviewer 3D HISTECH Ltd (RRID:SCR_017654; https://www.3dhistech.com/caseviewer), respectively.

## Immunohistochemistry

Deparaffinized sections or mouse brain cryosections were quenched for 10 min in 3% H$_2$O$_2$–10% (vol/vol) methanol. Antigen retrieval in deparaffinized sections was performed with a 10 mM citric acid solution at pH 6.0 at 95 °C for 20 min. Sections were rinsed 3 times in 0.1 M Tris-buffered saline (TBS) between each incubation period. Blocking for 1 h with 5% (vol/vol) normal goat or rabbit serum (NGS or NRS) (Vector Laboratories, #S-1000; RRID:AB_2336615 and #S-5000; RRID:AB_2336619) was followed by incubation with the primary antibody (Table S2) at 4 °C for 24 or 48 h in 2% (vol/vol) serum and with the corresponding biotinylated antibody (Vector Laboratories) or alkaline-phosphatase antibody (ab6722; RRID:AB_954595) for 1 h at room temperature. Sections were visualized by incubation with avidin-biotin-peroxidase complex (Thermo Fisher Scientific, ABC Peroxidase Standard Staining Kit #32020 or Ultra-Sensitive ABC Peroxidase Standard Staining Kit #32050), VectorSG Peroxidase Substrate Kit (Vector Laboratories, #SK-4700; RRID:AB_2314425) or ImmPACT Vector Red Substrate (Vector, #SK-5105; RRID:AB_2336524) and then mounted and coverslipped with DPX mounting medium (Sigma-Aldrich #06522). Bright-field section images were examined with Zeiss Imager.D1 microscope coupled to an AxioCam MRc camera and with Pannoramic 250 Flash III (3D Histech), and processed with ZEN 2011 software (Zeiss, Germany; RRID:SCR_013672; https://www.zeiss.com/microscopy/en/products/software/zeiss-zen.html) and Caseviewer 3D HISTECH Ltd (RRID:SCR_017654; https://www.3dhistech.com/caseviewer), respectively.

## Immunofluorescence

Deparaffinized sections were blocked with 5% (vol/vol) NGS and 0.1% (vol/vol) Triton X-100 (Sigma-Aldrich) in phosphate buffered saline (PBS) solution. Corresponding primary antibodies (Table S2) were incubated together overnight at 4 °C in 2% (vol/vol) NGS and adequate Alexa 488, 594, and 647-conjugated secondary antibodies (1:1000, Thermo Fisher Scientific; RRID:AB_2534117; RRID:AB_2534079; RRID:AB_2534073; RRID:AB_2536183), together with Hoechst 33342 (1:10000, Cell Signaling Technology; RRID:AB_10626776) to stain nuclei, were incubated simultaneously for 1 h at RT in 2% (vol/vol) serum. Sections were coverslipped using the DakoCytomation Fluorescent Mounting Medium (Dako, S302380-2). Immunofluorescence section images were examined using ZEISS LSM 980 with Airyscan 2 and processed with ZEN 2011 software (Zeiss, Germany; RRID:SCR_013672; https://www.zeiss.com/microscopy/en/products/software/zeiss-zen.html).

## Masson-Fontana

Staining of NM granules in 5 µm-thick paraffin-embedded human and mouse brain sections was performed using the Masson-Fontana Staining Kit (DiaPath; # 010228). This procedure is based on the ability of NM to chelate metals by reducing silver nitrate to a visible metallic state. Briefly, paraffin tissue sections were dewaxed and

rehydrated by heating at 60 °C for 10 min, followed by xylene (5 min, 3 times) and ethanol serial washes (100–95–70%-H$_2$O, 5 min each). Staining was performed by incubating the sections with ammoniac solution for 40 min at 56 °C, followed by sodium thiosulphate for 2 min and a final counterstain with Kernechtrot for 7 min. Between each step, samples were rinsed in distilled water.

### Nissl and hematoxylin-eosin staining
To perform the NM brain map, standard Nissl staining was performed for each animal in serial 30 μm-thick cryosections spanning the whole brain from the olfactory bulb to the beginning of the spinal cord. Areas from A16 to A12 were imaged with Zeiss Imager.D1 microscope coupled to an AxioCam MRc camera. Standard hematoxylin-eosin staining was performed in 5 μm-thick paraffin-embedded sections covering the SN/VTA and LC regions for each animal. In these sections, catecholaminergic neurons were identified by the visualization of unstained NM pigment.

### Intracellular NM quantification
Five μm-thick paraffin-embedded hematoxylin-eosin-stained sections or 30 μm-thick unstained cryopreserved sections representative of the different regions of interest (SN, VTA, LC, DVC) at different ages were selected from different animals ($n = 5$–8 tgNM per age) and all pigmented neurons per section were analyzed. Melanized catecholaminergic neurons were identified by the visualization of unstained NM brown pigment. H&E sections were scanned using the Pannoramic Midi II FL, HQ SCIENTIFIC ×60 and section images were acquired with CaseViewer software at an objective magnification of ×63. Alternatively, bright-field pictures from unstained sections were taken with the Zeiss Imager.D1 microscope coupled to an AxioCam MRc camera. All NM-positive neurons from each brain nuclei were analyzed by means of optical densitometry using ImageJ software (NIH, USA; RRID:SCR_003070; https://imagej.net/) to quantify the percentage (%) of cytosolic area occupied by NM pigment in hematoxylin-eosin-stained sections and the intracellular optical density (OD) of NM in unstained sections as previously reported[19]. For NM pigment OD measurements, the pixel brightness values for all individual NM-positive cells (excluding the nucleus) in all acquired images were measured and corrected for non-specific background staining by subtracting values obtained from the neuropil in the same images. All quantifications were performed by an investigator blinded to the experimental groups.

### Stereological cell counting
Assessment of (i) the total number of SNpc, VTA, LC and DVC TH-positive neurons, (ii) the number of NM-laden neurons, (iii) the total number of DA/NA neurons in the different regions of interest, (iv) the total number of ChAT-positive neurons in DVC, PPN and NBM, and (v) the total number of TPH-positive neurons in the DR, was performed according to the fractionator principle, using the MBF Bioscience StereoInvestigator 11 (64 bits) Software (Micro Brightfield; RRID:SCR_002526; https://www.mbfbioscience.com/products/stereo-investigator). Serial 30 μm-thick cryosections (every fourth section for SNpc, VTA, LC; every sixth section for PPN, NBM and DVC), or 5 μm-thick paraffin sections for the DR (every tenth section), covering the entire nuclei were included in the counting procedure. For SNpc, VTA, LC, PPN and NBM a 25% of the area was analyzed with the following sampling parameters: (i) a fixed counting frame with a width and length of 50 μm; (ii) a sampling grid size of 125 × 100 μm. The counting frames were placed randomly by the software at the intersections of the grid within the outlined structure of interest. For DR and DVC, the 100% of the area was analyzed. Cells in both brain hemispheres were counted following the unbiased sampling rule using a ×100 lens and included in the measurement when they came into focus within the dissector. A coefficient error of <0.10 was accepted. Data for the total numbers of TH-positive neurons, TPH-positive and ChAT-positive

neurons are expressed as absolute numbers. The total number of catecholaminergic SNpc, VTA, LC and DVC neurons was calculated by considering all TH$^+$NM$^+$, TH$^-$NM$^+$ and TH$^+$NM$^-$ neurons. The percentage of TH downregulation was calculated by considering the total number of TH$^+$NM$^+$ and the total number of TH$^-$NM$^+$ with respect to the total number of neurons containing NM in the different experimental groups. The number of tgNM and wt mice analyzed at each age is as follows: SNpc and VTA Young ($n = 7$ wt, $n = 7$ tgNM), Adult ($n = 13$ wt, $n = 6$ tgNM), Old ($n = 19$ wt, $n = 12$ tgNM); LC Pubertal ($n = 7$ wt, $n = 4$ tgNM), Young ($n = 5$ wt, $n = 6$ tgNM), Adult ($n = 14$ wt, $n = 5$ tgNM), Old ($n = 12$ wt, $n = 7$ tgNM); NBM Old ($n = 8$ wt, $n = 7$ tgNM); PPN Old ($n = 6$ wt, $n = 6$ tgNM); DR Old ($n = 6$ wt, $n = 5$ tgNM); DVC Young ($n = 5$ [TH]/6 [ChAT] wt, $n = 6$ tgNM), Adult ($n = 6$ wt, $n = 6$ tgNM), Old ($n = 6$ wt, $n = 6$ tgNM). All quantifications were performed by an investigator blinded to the experimental groups.

### Quantification of neuropathological parameters
The absolute number of extracellular NM (eNM) aggregates was estimated using the stereological parameters used for the stereological cell counting and in the same sections in which neurodegeneration was assessed (SNpc, VTA, LC and DVC). The number of p62-immunopositive Marinesco bodies and Lewy body-like inclusions were counted from SNpc, VTA, LC and DVC sections fluorescently immunostained with guinea pig anti-p62 (1:500, Progen #GP62-C), mouse anti-α-synuclein (1:500, BD Biosciences #610786) and TH (1:1000, Calbiochem #657012). The total number of p62-positive inclusions falling into each category was counted from one representative coronal section of the SNpc, VTA, LC and DVC nuclei in each animal. TgNM and wt mice were analyzed at different ages: 1 m ($n = 5$ for LC and DVC), 3 m ($n = 6$), 12 m ($n = 7$ for all regions), 20 m ($n = 6$ for LC and DVC, $n = 4$ for SN and VTA). All quantifications were performed by an investigator blinded to the experimental groups.

### Optical densitometry of neuronal fibers
The density of TH/DAT/VMAT2-positive fibers in the striatum, nucleus accumbens and olfactory tubercle was measured by densitometry in serial coronal sections covering the whole regions (4 sections/animal). TH-immunostained 30 μm-thick cryosections were scanned with an Epson Perfection v750 Pro scanner and the resulting images were quantified using ImageJ. Densitometry values were corrected for non-specific background staining by subtracting densitometric values obtained from the cortex. Data are expressed as optical density or absorbance defined by the logarithmic intensity of the light transmitted through the material using the formula: −log10 (Striatum Intensity/Cortex Intensity). TgNM and wt mice were analyzed at different ages: Adult TH ($n = 14$ wt, $n = 6$ tgNM), DAT ($n = 12$ wt, $n = 6$ tgNM), VMAT ($n = 7$–9 wt, $n = 4$–6 tgNM); Old TH ($n = 20$ wt, $n = 13$ tgNM), DAT ($n = 20$ wt, $n = 13$ tgNM), VMAT ($n = 12$–15 wt, $n = 13$–14 tgNM).

### Quantification of neuroinflammation parameters
Quantification of Iba-1 and GFAP-positive cells was performed in one SNpc, VTA, LC, and DVC representative section. Sections were scanned using the Panoramic Midi II FL, HQ SCIENTIFIC ×60 scanner and section images were acquired with CaseViewer software (RRID:SCR_017654; https://www.3dhistech.com/caseviewer). For quantification of Iba-1 and GFAP-positive cells, specific AI-based algorithms were implemented using the Aiforia 5.3 platform (RRID:SCR_022739, https://www.aiforia.com) as previously described[20]. Iba-1-positive cells were counted separately in two different groups according to their activation state: non-reactive (branched) and reactive (ameboid). GFAP-positive cells were counted individually. All quantifications were performed by an investigator blinded to the experimental groups. SN and VTA sections: Young ($n = 8$ wt, $n = 6$ tgNM), Adult ($n = 4$ wt, $n = 5$ [Iba-1]/6 [GFAP] tgNM), Old ($n = 6$ wt, $n = 5$ tgNM); LC sections: Young ($n = 8$ wt,

$n = 6$ tgNM), Adult ($n = 4$ wt, $n = 5$ tgNM), Old ($n = 5$ wt, $n = 4$ tgNM); DVC sections: Young ($n = 7$ [Iba-1]/8 [GFAP] wt, $n = 6$ tgNM), Adult ($n = 4$ wt, $n = 5$ tgNM), Old ($n = 4$ [Iba-1]/5 [GFAP] wt, $n = 4$ tgNM).

## Gene expression of human tyrosinase and dopaminergic markers

Total RNA was extracted from dissected ventral midbrain (vMB), comprising the SN and VTA regions, from young ($n = 4$ wt, $n = 5$ tgNM), adult ($n = 6$ wt, $n = 6$ tgNM) and old ($n = 6$ wt, $n = 12$ tgNM) mice. To assess whole-brain TYR expression, total RNA was also extracted from additional dissected brain regions (i.e., LC, OB, DVC, PFC, HIP) of adult mice ($n = 6$ wt, $n = 6$ tgNM). MirVana PARIS RNA and Native Protein Purification Kit (Thermo Fisher Scientific # AM1556) was used for total RNA extraction and RNA concentration was determined using a NanoDrop ND-1000 Spectrophotometer. $0.5\,\mu g$ of total RNA were retrotranscribed using High-Capacity cDNA Reverse Transcription Kit (Thermo Fisher Scientific # 4368814). qPCR was performed with 10 ng of cDNA per well in technical triplicates mixed with Taqman Gene Expression Master Mix (Applied Biosystems, # 4369016) and Taqman gene expression assays (human tyrosinase [TYR] Hs00165976_m1; mouse TH Mm00447557_m1; Slc6a3 [DAT] Mm00438388_m1; Slc18a2 [VMAT2] Mm00553058_m1, Applied Biosystems) using standard procedures in a 7900HT Fast Real Time Instrument (Applied Biosystems). Threshold cycles (Cts) for each target gene were normalized to the geometric mean of three endogenous reference genes (Gapdh Mm99999915_g1, Rpl19 Mm02601633_g1 and Ppia Mm02342430_g1). Water was included in the reaction as a non-template (negative) control. The relative expression was calculated with the ΔCt-method.

## Immunoblot

Dissected SN-VTA from adult ($N = 6$ wt, $N = 8$ tgNM) and Str tissue from old ($N = 6$ wt, $N = 8$ tgNM) mice were homogenized in RIPA buffer supplemented with protease inhibitors (Roche) and cell extracts clarified by centrifugation at $13,000 \times g$ for 30 min at 4 °C. Protein concentrations were quantified using the BCA method and subjected to SDS-PAGE. Proteins were resolved in 10 or 15 % polyacrylamide gels and transferred onto $0.45\,\mu m$ nitrocellulose membranes (Amersham). Blocking with 5% milk powder in PBS was followed by overnight incubation at 4 °C with the primary antibodies diluted as indicated in Table S2. Incubation with the secondary antibodies goat anti-rat (#NA935; RRID:AB_772207), donkey anti-rabbit (#NA934; RRID:AB_772206) and sheep anti-mouse (#NXA931; RRID:AB_772209) (all 1:1000, from Amersham), was performed for 1 h at RT. Band densitometry, normalized to β-actin expression, was measured using ImageJ image analysis software (RRID:SCR_003070; https://imagej.net/).

## In vivo microdialysis

To assess local effects of nomifensine (DAT and NET inhibitor) on striatal DA release, extracellular DA concentration was measured by in vivo microdialysis experiments as previously described[93,94]. Briefly, the drug was dissolved in artificial cerebrospinal fluid (aCSF in mM: NaCl, 125; KCl, 2.5; CaCl$_2$, 1.26 and MgCl$_2$ 1.18) and administered by reverse dialysis at 10 and 50 μM (uncorrected for membrane recovery). Concentrated solutions (1 mM; pH adjusted to 6.5–7 with NaHCO$_3$ when necessary) were stored at −80 °C and working solutions were prepared daily by dilution in aCSF. One concentric dialysis probe (Cuprophan membrane; 6000 Da molecular weight cut-off; 1.5 mm-long) was implanted in the striatum (AP, 0.5; ML, −1.7; DV, −4.5 in mm) of isoflurane-anesthetized mice (wt and tgNM, $n = 7$ mice per group). Microdialysis experiments were performed in freely-moving mice 24 h after surgery. Probes were perfused with aCSF at 1.5 μL/min. Following an initial 100-min stabilization period, 5 or 7 baseline samples were collected (20 min each) before local drug application by reverse dialysis and then successive dialysate samples were collected. The concentration of DA in dialysate samples was determined by HPLC

coupled to electrochemical detection (+0.7 V, Waters 2465), with 3-fmol detection limit. The mobile phase containing 0.15 M NaH$_2$PO$_4$.H$_2$O, 0.9 mM PICB8, 0.5 mM EDTA (pH 2.8 adjusted with orthophosphoric acid), and 10% methanol was pumped at 1 ml/min (Waters 515 HPLC pump). DA was separated on a 2.6 mm particle size C18 column (7.5 × 0.46 cm, Kinetex, Phenomenex) at 28 °C. Microdialysis data are expressed as femtomoles per fraction (uncorrected for recovery) and are shown in figures as percentages of basal values (individual means of 5–7 pre-drug fractions).

## UPLC-MS/MS

Chromatographic determination of catecholaminergic, serotonergic and cholinergic metabolites in brain and peripheral samples was performed by UPLC-MS/MS using our previously validated method[67] with modifications as follows: (1) *Brain and peripheral samples collection*. Heart, intestines and several brain regions including vMB, LC, Str, PFC, DR-PPN and DVC were dissected, frozen on dry ice, and stored at −80 °C until use; (2) *Sample preparation*. The day of analysis, samples were homogenized with 300 μl of 250 mM formic acid (FA) and split in two: (i) 55 μl for acetylcholine (Ach) determination, which were diluted 1:4 with 0.1% formic acid in acetonitrile containing 100 nM of Acetylcholine-d4 Chloride as internal standard (IS); (ii) 240 μl for catecholaminergic and serotonergic determination to which a mixture (500 nM each) of dopamine-d$_4$ hydrochloride (DA-d$_4$) and serotonin-d4 Hydrochloride (5-HT-d4) were added as IS. After centrifugation, supernatants were filtered using an Ostro™ protein precipitation and phospholipid removal plate (Waters, USA) prior the injection in the UPLC-MS/MS system. On the other hand, the pellet of the 240 μl sample was further processed to determine the "protein-bound" (PB) fraction of 5-*S*-cysteinyldopa (5SCD) and 5-*S*-cysteinyldopamine (5SCDA). 5SCD and 5SCDA standards were kindly provided by Profs. Kazumasa Wakamatsu and Shosuke Ito at the Fujita Health University, Aichi, Japan. Aminochrome standard (0.5 mM) was freshly prepared as previously described[67,95]; (3) *Reductive hydrolysis and alumina extraction of catecholic compounds*. Preparation of the PB fraction samples was performed using the reductive hydrolysis and alumina extraction method of Murakami et al.[96] with some modifications. The pellet was washed with 1 ml of a 1:1 mixture of methanol and chloroform, vortex-mixed and centrifuged at $20,000 \times g$ for 10 min at 4 °C. Three hundred μl of a mixture containing 6 M HCl, 5% thioglicolic acid and 1% phenol were added to the resulting pellet into a sealed-capped tube. Tubes were then purged with a stream of nitrogen and incubated for 16 h at 110 °C. After cooling, 100 μl of the hydrolysate were added to a tube containing 50 mg of acid-washed alumina prior the addition of 200 μl of 1% Na2S2O5 – 1% EDTA.2Na and 500 μl 2.7 M Tris. HCl – 2% EDTA.2Na (pH 9.0). Tubes were vigorously mixed on a microtube mixer for 5 min and then centrifuged at $20,000 \times g$ for 10 min at 4 °C. After removal of the supernatant, the alumina was washed with 1 ml of MilliQ water and centrifuged at $20,000 \times g$ for 10 min at 4 °C three times. 5SCD and 5SCDA were finally eluted from alumina with 100 μl of 0.4 M HClO$_4$ by shaking for 2 min on a microtube mixer. Seven μl were injected into the UPLC-MS/MS system under MIX3 conditions; (4) *UPLC-MS/MS analysis*. The chromatographic separation of samples for ACh determination was performed on a Cortecs UPLC HILIC (1.6 μm; 2,1 × 75 mm) column coupled to a Cortecs UPLC HILIC VanGuard pre-column (Waters). Column temperature was set at 50 °C and samples were maintained at 6 °C in the thermostatic autosampler. The mobile phase consisted of solvent A (Acetonitrile + 0.1% FA) and solvent B (10 mM ammonium acetate in MilliQ water) at a flow of 0.5 mL/min with isocratic 70% A- 30% B conditions during 2.2 min. Samples for catecholaminergic and serotonergic determination were injected five times into the UPLC-MS/MS system to analyze different sets of compounds, i.e., MIX1, MIX2, MIX3, MIX3-PB and MIX4. MIX1 includes dopamine (DA), noradrenaline (NA), 3-methoxytyramine (3MT), 3,4-dihydroxyphenylalanine (L-DOPA) and aminochrome (AC); MIX2

includes 3,4-Dihydroxyphenylacetic acid (DOPAC), 3,4-dihydroxymandelic acid (DOMA) and vanillylmandelic acid (VMA); MIX3 and MIX3-PB include 5SCD and 5SCDA; MIX4 includes serotonin (5-HT), tryptophan (Trp) and 5-hydroxyindole-3-acetic acid (5-HIAA). An Acquity HSS T3 (1.8 μm, 2.1 mm × 100 mm) column coupled to an Acquity HSS T3 VanGuard (100 Å, 1.8 μm, 2.1 mm × 5 mm) pre-column was used to detect MIX1-3 analytes while an Acquity UPLC BEH C18 (1.7 μm 2.1 × 100 mm) column coupled to an Acquity BEH C18 1.7 μM VanGuard pre-column was used to detect MIX4. Column temperature was set at 45 °C for MIX1-3 and at 55 °C for MIX4. The mobile phase consisted of solvent A (methanol 100%) and solvent B (25 mM FA in MilliQ water) at a flow of 0.4 ml/min (0.5 ml/min for MIX4) with the following gradient profiles: (i) MIX1 and MIX2: 0.5 % B maintained for 0.5 min, 5 % B at 0.9 min maintained for 2.1 min, 50 % B at 2.8 min maintained for 1.2 min, 0.5 % B at 4.1 min maintained 0.2 min for equilibration; (ii) MIX3 and MIX3-PB: 0.5 % B maintained for 0.5 min, 8 % B at 2.6 min, 50 % B at 2.9 min and maintained for 0.6 min, 0.5 % B at 3.7 min maintained 0.2 min for equilibration and (iii) MIX4: 1% B maintained for 0.5 min, 25 % B at 3 min, 50 % B at 3.1 min and maintained for 0.5 min, 1 % B at 3.6 min maintained 0.4 min for equilibration. The mass spectrometer detector operated under the following parameters: source temperature 150 °C, temperature 450 °C, cone gas flow 50 L/hr, desolvation gas flow 1100 L/hr and collision gas flow 0.15 ml/min. Argon was used as the collision gas. The capillary voltage was set at: 0.5 kV (MIX1, MIX3 and MIX3-PB), 2 kV (MIX2) or 3 kV (MIX4, ACh). The electrospray ionization source was operated in both positive and negative modes, depending on the analyte. Multiple Reaction Monitoring (MRM) acquisition settings for the targeted metabolites are summarized in Table S3. Samples with a concentration between the limit of detection (LOD) and limit of quantification (LOQ) or bigger than LOQ were considered acceptable; samples with a concentration lower than LOD were considered as the LOD value. Catechol oxidation was measured using the formula AC + 5SCDA + 5SCDA-PB/DA + 5 SCD + 5SCD-PB/L-DOPA. DA synthesis was measured using the formula DA + NE + DOMA + VMA + 3MT + DOPAC /L- DOPA. DA degradation was measured using the formula DOPAC + 3MT/DA. NA synthesis was measured using the formula NA/DA. NA degradation was measured using the formula DOMA + VMA/NA. Data was normalized by the protein concentration (determined by BCA) and presented as the percentage of the wt concentration or ratio. SN-VTA Young ($n = 6$ wt, $n = 13$ tgNM), SN-VTA Old ($n = 6$ wt, $n = 8$ tgNM), Str Young ($n = 9$ wt, $n = 13$ tgNM), Str Old ($n = 6$ wt, $n = 11$ tgNM), LC Young ($n = 9$ wt, $n = 13$ tgNM), LC Old ($n = 6$ wt, $n = 8$ tgNM), PFC Young ($n = 8$ wt, $n = 13$ tgNM), PFC Old ($n = 6$ wt, $n = 11$ tgNM), DVC Young ($n = 8$ wt, $n = 13$ tgNM), DVC Old ($n = 6$ wt, $n = 12$ tgNM), heart Young ($n = 9$ wt, $n = 13$ tgNM), heart Old ($n = 11$ wt, $n = 12$ tgNM), intestines Young ($n = 9$ wt, $n = 13$ tgNM), intestines Old ($n = 17$ wt, $n = 17$ tgNM), PPN-DR Young ($n = 8$ wt, $n = 12$ tgNM), PPN-DR Old ($n = 4$ wt, $n = 10$ tgNM).

## Behavioral analyses

All behavioral analyses were performed at the same time of the day (9 am−1 pm): (1) Beam test. Mice were placed at the beginning of an elevated horizontal beam bar (40 cm). The time it took the mice to cross the beam was measured for a maximum of 120 seconds. When an animal fell, the maximum time of 120 s was assigned. Animals that failed at doing the task (i.e., not crossing or going backwards) were removed from the analysis. Young ($n = 15$ wt, $n = 13$ tgNM), Adult ($n = 14$ wt, $n = 17$ tgNM), Old ($n = 13$ wt, $n = 10$ tgNM); (2) Habituation and dishabituation (olfaction test). Three different cotton swabs were presented to mice (5 min each), the first one was presented for object habituation, the second one was impregnated with water and the third one was impregnated with lemon essence (Essenciales). The number of times the animal went towards the second and third cotton swabs and the time the animal spent sniffing them were measured. The discrimination index (DI) was calculated according to the formula: (Time

exploring lemon essence−Time exploring water)/(Time exploring lemon essence+Time exploring water). Young ($n = 15$ wt, $n = 15$ tgNM), Adult ($n = 18$ wt, $n = 17$ tgNM), Old ($n = 13$ wt, $n = 10$ tgNM); (3) Grip strength test. Mice were held by the middle/base of the tail and allowed to grasp a series of increasing weights consisting of tangled fine gauge stainless steel wire attached to steel chain (13.2, 19.7, 25.9, 32.1, 38.4, 44.6 g). Mice were then lifted carrying the corresponding weight with their forepaws. Time holding each weight (for a total of 5 s each) was assessed, being 30 seconds the maximum time when succeeding in holding all weights. Grip latency (s) was calculated as a sum of the time holding the increasing weights. Young ($n = 17$ wt, $n = 24$ tgNM), Adult ($n = 19$ wt, $n = 19$ tgNM), Old ($n = 18$ wt, $n = 12$ tgNM); (4) Novel Object Recognition Test. This protocol is used to evaluate cognition in mice, particularly recognition memory. It assesses inherent behavior in mice to explore a novel object. This test consisted in exposing the animals to objects that with habituation become familiar and after 24 h one of the object was changed with a new one and the discrimintation index ([time exploring the new object]-[time exploring the familiar object]/total time) was measured. Recorded data was analyzed by the Applied Research in Laboratory Animals Platform staff at Parc Científic de Barcelona (Spain). Adult ($n = 18$ wt, $n = 17$ tgNM), Old ($n = 14$ wt, $n = 10$ tgNM); (5) Step down test. The passive avoidance or step-down test [Passive Avoidance - Step Down for Mice (vibrating platform), #40570, Ugo Basile] consists of a platform located in a controllable electrified net. During training, mice were placed in the platform and received an electric shock of 0.3 mA when stepping down the platform. After 24 hours, mice were placed again in the same platform and the latency of stepping down was measured, though this second time the net was not electrified. The time on the platform was calculated by subtracting the latency time from the first day to that of the second day to normalize for the inter-individual variability in curiosity/activity of each mouse. Adult ($n = 17$ wt, $n = 15$ tgNM), Old ($n = 13$ wt, $n = 10$ tgNM); (6) Open field. Mice were placed in a square open field arena of 80 × 80 cm of white methacrylate, brightly lit (300 lux), where the center and the periphery can be distinguished using a tracking software (SMART 3.0 Panlab-Harvard Apparatus). Recorded data was analyzed by the Applied Research in Laboratory Animals Platform staff at Parc Científic de Barcelona (Spain). Adult ($n = 18$ wt, $n = 17$ tgNM), Old ($n = 14$ wt, $n = 9$ tgNM); (7) Polysomnography. Eight months aged male tgNM mice and their wt littermates were conventionally prepared under anesthesia (ketamine and xylazine, 100/10 mg/kg. i.p.) for polysomnography. Briefly, 3 electrodes were screwed to skull above parietal and frontal cortex and cerebellum as the reference for electroencephalogram (EEG) whereas 2 wire electrodes were slipped between neck muscles for electromyogram (EMG). Leads were then brazed to a miniature plug (Plastics One, Bilaney, Germany), cemented to skull using acrylic cement (Superbond, C&B Sun Medical) and coated using dental cement (Paladur, Heraeus Kulzer). A subcutaneous injection of carprofen (5 mg/kg) was administered for pain caring. Once recovered, mice were housed in individual barrels under 12 h light/dark cycle (light-on 7 a.m.) and constant temperature, tethered to the acquisition system and recorded 1 week out of 3 over 12 consecutive months until natural death. Based on amplified/digitized unipolar EEG and bipolar EMG signals, vigilance states were visually scored using a 5 s sliding window frame and assigned as Waking (W), Slow Wave Sleep (SWS) and Paradoxical Sleep (PS) to produce hypnograms and state quantifications (Sleepscore, Viewpoint). Adult ($n = 7$ wt, $n = 8$ tgNM), Old ($n = 6$ wt, $n = 5$ tgNM); (8) Tail suspension test. This test was performed as previously described[98]. Briefly, animals were suspended by their tails with tape into a suspension bar. To avoid the tail climbing behavior, a 2 cm methacrylate tube was passed through the tail before suspending the animal. The escape-oriented behaviors (i.e., fore and hind limbs movement) were quantified during six minutes. Data is presented as total immobilization time (s). Young ($n = 16$ wt, $n = 16$ tgNM), Adult ($n = 15$ wt, $n = 13$ tgNM), Old ($n = 11$ wt, $n = 11$ tgNM); (9) Vocalizations

test. Animals that vocalized during the 6 min of the tail suspension test were registered with a yes/no score. Young ($n = 16$ wt, $n = 16$ tgNM), Adult ($n = 21$ wt, $n = 19$ tgNM), Old ($n = 18$ wt, $n = 12$ tgNM). Behavioral equipment was cleaned with 70% ethanol after each test session to avoid olfactory cues. All behavioral tests were performed by an investigator blinded to the experimental groups.

### Assessment of peripheral functions

(1) *Heart Rate*. The PhysioSuite apparatus (Kent Scientific) was used to measure the heart rate under mild anesthesia. The isoflurane rate used was 1.5% and 1 l/min $O_2$. The clip sensor was located in the right hind paw for 30 seconds before starting the recording and the first 10 recordings were taken and represented as average beats/minute. Young ($n = 12$ wt, $n = 13$ tgNM), Adult ($n = 12$ wt, $n = 11$ tgNM), Old ($n = 12$ wt, $n = 22$ tgNM). (2) *Blood pressure*. Blood pressure was measured in a non-invasive way using the tail-cuff method (LE-5002 Non-Invasive Blood Pressure Meter, Panlab). The systolic and diastolic blood pressure (in mmHg) were assessed in two independent sessions for each mouse and the results were calculated as the mean from the valid values of 10 measurements in each session. $n = 19$ wt, $n = 15$ tgNM; (3) *Respiratory rate*. Respiration frequency was determined using microCT measurements acquired with Quantum FX imaging system (Perkin Elmer. 940 Winter St. Waltham, Massachusetts. EEUU), specifically designed for small lab animals. For the scanning procedure, animals were anesthetized with isoflurane (5% induction phase, 1% maintenance). The air flow was set to 0.8 l/minute. Defined parameters for image acquisitions were: field of view 73 mm, acquisition time 2 minutes, current voltage 70 mV and amperage 200 uA. Image reconstruction was based on Feldkamp´s method. Imaging data was analyzed by the Preclinical Imaging Platform staff at Vall d´Hebron Research Institute (Spain). $n = 12$ wt, $n = 13$ tgNM. (4) *Intestinal transit time*. Mice were administered a red dye (Carmine red dye, Sigma #C1022) in 0,5% methylcellulose (Sigma #M7027) by oral gavage. Animals were individualized in new cages without bedding and monitored for a total time of 4 hours. The time period until the animal defecated the first dyed feces was registered. Adult ($n = 19$ wt, $n = 18$ tgNM), Old ($n = 18$ wt, $n = 12$ tgNM). All tests were performed by an investigator blinded to the experimental groups.

### Statistics

Statistical analyses were performed using GraphPad Prism v6 software (RRID:SCR_002798; http://www.graphpad.com/) using the appropriate statistical tests, as indicated for each figure legend in the *Source Data* file. We used the sample size calculator GRanmo (https://www.datarus.eu/aplicaciones/granmo/) to calculate sample size and sample sizes are equivalent to those reported in previous similar publications[19]. Outlier values in qPCR experiments were identified by ROUT (Q = 1.0%) test and excluded from the analyses when applicable. Outlier values in UPLC experiments were identified by the Grubbs' test (i.e., Extreme Studentized Deviate, ESD, method) and excluded from the analyses when applicable. All data is represented as box-and-whisker plots (median, minimum, maximum and interquartile range). Since all experiments had a relatively small number of mice ($n = 4$–24 mice/group), nonparametric tests were used because a Gaussian distribution could not be assumed considering sample size. In all analyses, the null hypothesis was rejected at the 0.05 level.

### Reporting summary

Further information on research design is available in the Nature Portfolio Reporting Summary linked to this article.

## Data availability

All raw data generated in this study are provided in the Supplementary Information/Source Data file and are available in the ZENODO database under accession code https://doi.org/10.5281/zenodo.11355658. The latter also includes the raw UPLC-MS/MS data. All detailed protocols have been uploaded in the public repository Protocols.io under the accession code: https://doi.org/10.17504/protocols.io.4r3l2qq1ql1y/v3. Further information and requests for resources and reagents should be directed to and will be fulfilled by the Lead Contact, Miquel Vila (miquel.vila@vhir.org). Source data are provided with this paper.

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

## Acknowledgements

This research was funded by: Aligning Science Across Parkinson's through The Michael J. Fox Foundation for Parkinson's Research, USA (ASAP-020505 to M.V.); The Michael J. Fox Foundation for Parkinson's Research, USA (MJFF-007184 and MJFF-001059 to M.V.); Ministry of Science and Innovation (MICINN), Spain (PID2022-141700OB-I00, MCIN/ AEI/10.13039/501100011033 to A.B. and PID2020-116339RB-I00 to M.V.); EU Joint Program Neurodegenerative Disease Research (JPND), Instituto de Salud Carlos III, EU/Spain (AC20/00121 to MV): La Caixa Bank Foundation, Spain (INPhINIT fellowship, code LCF/BQ/DI18/1166063 to J.C.; Junior Leader Fellowship LCF/BQ/PR19/11700005 to A.L.; Health Research Grant, ID 100010434 under the agreement LCF/PR/HR17/ 52150003 to MV); Parkinson's U.K. (to M.V.); Research grants from Association France Parkinson and Fondation Neurodis (to P.F.); China Scholarship Council (CSC; no. 202008620080) to J.L.; Ministry of Economy and competitiveness (MINECO), Spain with co-funding from FEDER (E.U.) (SAF2015-73997-JIN to A.L.); Ministry of Economy and competitiveness (MINECO), Spain (BES-2017-080191 to N.P.). We are grateful to: the Neurological Tissue Bank of the Biobanc-Hospital Clinic-IDIBAPS (Barcelona, Spain) and the Biobanco en Red de la Región de Murcia-BIOBANC-MUR (Murcia, Spain) for human sample procurement; Sandra Lope-Piedrafita (UAB, Barcelona, Spain) for technical assistance in the performance of MRI experiments in rodents; the Mesoscopic Imaging Facility (MIF) at the European Molecular Biology Laboratory (EMBL) (Barcelona) for support in brain clarification procedures; Sandra Mancilla from the Unit 20 of CIBER in Bioengineering, Biomaterials & Nanomedicne (CIBER-BBN) at VHIR (Barcelona, Spain) and Beatriz Rodríguez-Galván (VHIR) for assistance in histological processing; Ignasi Sahún from the Phenotyping Unit at Parc Científic de Barcelona (Barcelona, Spain), Guillem Colell and Anna Pujol from the Laboratory Animal Service at VHIR, Lidia Castillo-Mariqueo and Daniela Marin-Pardo from the Translational Behavioral Neuroscience Lab at UAB and Pamela Dominguez at VHIR for assistance in animal behavior; Eliana Markidi for

immunoblot analyses and laboratory assistance; Alex Rovira (Neuror-adiology section, Vall d'Hebron Hospital, Barcelona, Spain) for providing the human NM-MRI images; Ellen Gelpi (Neurological Tissue Bank, Hospital Clínic-IDIBAPS, Barcelona, Spain & Institute of Neurology, Medical University of Vienna, Austria) for providing the human autopsy images; Dr. Takafumi Hasegawa (Department of Neurology, Tohoku University School of Medicine, Sendai, Japan) for kindly providing the TYR plasmid; Dr. Jin Son and Dr. José López-Barneo from Hospital Virgen del Rocío (Sevilla, Spain) for kindly providing the rat TH promoter plasmid; Prof. Kazumasa Wakamatsu and Prof. Shosuke Ito (Fujita Health University, Aichi, Japan) for kindly donating the 5SCD and 5SCDA standards; Professor Gary Miller (Columbia University, USA) for kindly donating the VMAT2 antibody. All diagrams and schematic experimental plans were created with BioRender.com. For the purpose of open access, the author has applied a CC-BY 4.0 public copyright license to all Author Accepted Manuscripts arising from this submission.

## Author contributions

M.V. conceived the original idea for the study; A.L. and M.V. planned and oversaw all aspects of the study; A.L., N.P., and M.V. performed and analyzed most of the experiments. M.G.-S. performed and analyzed UPLC-MS/MS experiments; A.N. quantified and analyzed intracellular inclusions; S.A., J.L., and P.F. performed and analyzed sleep-related studies; C.G.-S. helped quantify neurons by stereology; M.L.-P. and L.G.-Ll. performed and analyzed behavioral tests; J.C. quantified neuroinflammation parameters; L.l., M.-R., and A.B. performed and analyzed in vivo microdialysis experiments; H.X. contributed to statistical analyses; A.P., T.C., and J.R.-G. provided technical support to most of the experiments; G.P. performed and analyzed the PPN- and NBM-related experiments; I.C.C. helped generate and expand the initial TgNM mouse colony; A.L., N.P., and M.V. wrote the manuscript with input and substantial revisions from all authors.

## Competing interests

The authors declare no competing interests.

## Additional information

[1]Neurodegenerative Diseases Research Group, Vall d'Hebron Research Institute (VHIR)-Network Center for Biomedical Research in Neurodegenerative Diseases (CIBERNED), 08035 Barcelona, Spain. [2]Aligning Science Across Parkinson's (ASAP) Collaborative Research Network, Chevy Chase, MD 20815, USA. [3]Institut de Neurociències-Autonomous University of Barcelona (INc-UAB), 08193 Cerdanyola del Vallès, Spain. [4]CNRS UMR5292, INSERM U1028, Lyon Neuroscience Research Centre (CRNL), SLEEP team "Physiopathologie des réseaux neuronaux responsables du cycle veille-sommeil", Lyon, France. [5]University Claude Bernard, Lyon 1, Lyon, France. [6]Department of Neuroscience and Experimental Therapeutics, Institute of Biomedical Research of Barcelona (IIBB), Spanish National Research Council (CSIC); Center for Networked Biomedical Research on Mental Health (CIBERSAM), 08036 Barcelona, Spain. [7]Systems Neuropharmacology Research Group, Fundació de Recerca Clínic Barcelona-Institut d'Investigacions Biomèdiques August Pi Sunyer (FRCB-IDIBAPS), 08036 Barcelona, Spain. [8]Department of Psychiatry and Forensic Medicine-Autonomous University of Barcelona (INc-UAB), 08193 Cerdanyola del Vallès, Spain. [9]Department of Biochemistry and Molecular Biology, Autonomous University of Barcelona, 08193 Barcelona, Spain. [10]Catalan Institution for Research and Advanced Studies (ICREA), 08010 Barcelona, Spain. [11]These authors contributed equally: Ariadna Laguna, Núria Peñuelas. ✉e-mail: miquel.vila@vhir.org

