## [Peer Review File · Nature Communications]

Modelling human neuronal catecholaminergic pigmentation in rodents recapitulates age-related neurodegenerative deficitsREVIEWER COMMENTS

Reviewer #1 (Remarks to the Author):

In this study, Laguna and colleagues extended their previous work on the important connection between neuromelanin (NM) and neurodegeneration. Neuromelanin is absent in rodents and this is considered to be major limitation in modeling Parkinson's disease, where neuromelanin-rich neurons seem to preferentially degenerate. Here, the authors report the development of a tissue-specific transgenic mouse model based on the TH promoter-regulated expression of the enzyme tyrosinase, a key player in neuromelanin synthesis. The manuscript is essentially descriptive, but provides an important description of a model that may prove to be very useful for the field of PD research, given the existing limitations of the models available. With a few adjustments, I believe the study will be of interest for the broad scientific community.

Specific comments

As the authors mention, not all aged people develop PD/neurodegeneration. Presumably, they all have NM. The explanation that all humans would have incipient degeneration seems weak, as age-associated frailty can arise for many reasons besides neurodegeneration. An associated question is why would all mice show signs of disease?

And why would there be NM in humans and not in rodents?

Perhaps the authors can discuss this

What is the prevalence of eNM granules in the tg animals vs normal and PD brains? Can the authors include this comparison?

Table S1 relates to data on Fig 2. What age is described in Table S1? It seems that in older animals A6 may have more NM. Is this significant?

Can the authors further characterize the LB-like inclusions in Fig 4 in order to support they are indeed LB-like? What about pS129 α -syn? Fibrillar α -syn antibodies? Ubiquitin? Recent LB-markers including mitochondrial remnants?

Without detailed characterization the authors should not refer to the structures observed as LBs (e.g. in line 180).

Do the animals show SAA-positivity in CSF/skin/plasma?

Formation of LBs in human brain is not limited to melanized neurons (as cortical brain areas in DLB). The authors should consider this as it suggests NM is not necessary for LB formation.

PD is more prevalent in males than in females. Did the authors observe gender differences? I suspect not, but perhaps the authors can show data to clarify this issue.

Reviewer #2 (Remarks to the Author):

Comments to the authors:

The manuscript “ Modelling human brain-wide pigmentation in rodents recapitulates age-related multisystem neurodegenerative deficits” by Laguna et al. generated a tissue-specific transgenic mouse (termed tgNM) that mimics the human age-dependent brain-wide distribution of neuromelanin (NM) within catecholaminergic regions and these animals display age-related neuronal dysfunction and degeneration affecting numerous brain circuits as well as peripheral organs, linked to motor and non-motor deficits, reminiscent of early neurodegenerative stages.

The group has validated the neuronal pigment induced by TYR overexpression in rodent catecholaminergic neurons closely resembles human NM. This supports the relevance of their approach to investigate the impact of human-like NM pigment on age-dependent neuronal function and viability in vivo.

The tgNM mouse model marks a significant milestone in the study of age-dependent NM accumulation ´s potential influence on neuronal function and viability. This model offers a unique opportunity to address these questions with a context highly relevant to humans. These animals recapitulate the biological and temporal complexity of age-dependent neurodegeneration, affecting various brain neurotransmission systems and the body leading to motor and non-motor symptoms. The authors have found there were also the alpha-synuclein pathology and neuroinflammatory changes in this tgNM mouse model which is relevant to human non-diseased aged and PD brains. Those changes may start prior to evident neurodegeneration and be associated to the presence of incipient eNM released from early degenerating neurons. Even though not all the changes were dramatic, these animals could be used to combine with additional disease relevant genetic or environmental manipulations. Furthermore, these animals could also be used to test potential therapeutic strategies in a human disease state-informed context at different stages of the disease, which should facilitate the translation of findings to humans.

The work is really significance to the field. This model may open new research avenues in the field of brain aging and neurodegeneration.

However, there are some parts can be improved. So, the manuscript needs to have some major revisions to be published.

Major points:

1. For Statistics, the majority was t-test has been used to compare the WT and tgNM mice, so it is a pity that the age differences were not compared in most of the figures.
2. In consistency of age groups, especially related to LC pathology, including figure 5 and S7, different ages group have been presented in different data sets.
3. The discussion part is quite general, more summarized the previous work, not so much on the result from the paper.
4. ISH methods missing for figure S1C.
5. For DVC, TYR gene expression level is relatively low (figure S1A), but intracellular NM density is

quite high (A2 in figure 2B), how to explain that?

Minor points:

1. Line 210-211 “An in-depth analysis of PS evidenced significantly reduced numbers of bouts concomitant to a significant increase of their duration in both adult and old tgNM mice (Figure 5C)”. it is not correct since for the duration in adult was not significant.
2. The vocalization was recorded during the 6 min of the tail suspension test, why the number of mice were different between TST and vocalization?
3. Method part, line 203-204, “Data are expressed as optical density or absorbance defined by the logarithmic intensity of the light transmitted through the material using the formula: $-\log_{10}(\text{Striatum Intensity}/\text{Cortex Intensity})$ ”. Which data had been analyzed by the latter way? And where was the data for the old mice since it mentioned there was old mice in line 206-207?
4. There was no number of mice for Novel Object Recognition Test.
5. Volume and page information were missing for “Vila, M. (2019). Neuromelanin, aging, and neuronal vulnerability in Parkinson's disease. *Movement disorders : official journal of the Movement Disorder Society.*”
6. TH antibody, 1:3000-1:20000 depending on the animal age, which method was used? The same for anti-ChAT.
7. No information for ProCathepsin antibody
8. Were LC3 I and II from the same gel and blotting?

Points for figures:

Figure 1

There was no H&E and M-F staining for wt mice, the figure legends should be modified.

Figure 3

A, wt adult, there must be some point(s) missing.

B, the Y axis of the figure is not correct, not second.

E What could be the reason for TH positive cells of old mice increased instead of decreased?

F, what were the grey square and circle stand for? Was there no data for young mice?

The text in result parts showed that 3F could explain the result from E, but the comparison was different, E was the comparison between the WT and tgNM, F only show the age difference, how to conclude E from F?

G, the Y axis is not correct, should be total NM containing neuron from the text, figure legend is also not correct.

Figure 4

C, What are the white circles and squares stand for?

Figure 5

A, The label of Y axis was confusing.

C, “s, seconds” should be added in the Y axis for the duration.

ABC, no data for young mice, since LC related to non-motor symptom and changes already at PEBERTAL, behavioral tests should be done at young age to show the early onset of non-motor symptoms.

F, no complete data for all four ages, figure legend was not precise.

Figure 6

C “cortex” in the result section, “PFC” in the figure legend while “Ctx” in the figure, should be consistent.

Figure 8

F, Ach was not mentioned in the result part.

Figure S1

A, there was significant difference between LC and DVC which was not indicated in the figure.

B, It was mentioned in Figure legend that “In B; *p<0.05” but there was no “*” in the figure

How was the value of X and Y axis calculated?

What is the negative control of C?

Figure S4

A, were there individual dots missing for Th and Dat for Young mice? Were there only three samples for some of the group?

C, the Y axis of the quantification is not proper.

S4C and S4D using different ages of mice,

S4D, statistics was specifically motioned, was it different from all the others? The “*” was wrongly put in the figure. It will be good to have the image of the immunostaining.

Figure S5

no adult data even though figure 3A indicated that all the data had been collected for all the age groups.

A, “SN” should be “SN-VTA”

D, the result of statistics was not mentioned.

Figure S6

The title of figure legend for S6 was wrong.

A it seems that the quantification has been normalized to WT, hasn't it?

B not correctly cited in the text and no result.

Figure S7

B, It is a bit strange that the speed of old mice was higher than adult.

F, What are the white circles and squares stand for?

Figure S8

B, The structures were indicated for UPLC right was not completely right. “serotonergic” should be added in the figure legend.

Figure S9

A, spelling for histology was wrong.

D, “DNV” should be “DVC”.

Reviewer #3 (Remarks to the Author):

Comments to the authors :

This study is an original and innovative work. The authors evaluated and validated a new mouse model that better mimics human neuropathologies implying neuromelanin like in Parkinson's disease. This work should open new ways to study neurodegenerative diseases as well as to test the efficacy of drug treatment. The authors combined different methods (behavioral testing, immunostaining, MRI, ...) to assess and explain the evolution of the brain pigmentation with age in this new mouse model.

General questions :

- In this study, did you use male or female mice ? Can you explain your choice. If you used males and females, can you precise the male/female ratio ?

- Did the tgNM mice present the same response to anesthesia (isoflurane) compared to wt ? In particular for microdialysis, peripheral functions assessment and MRI experiments ?

More specific questions :

#1; L67 : I do not understand why you mentioned an "age-related loss of pigmented SNpc and LC neurons" in healthy aging brain whereas the 1st sentence in your introduction mentioned that "Humans progressively accumulate with age the dark-brown pigment neuromelanin (NM) within catecholaminergic brain nuclei".

#2; L107 : Why did you choose to perform this experiment at 12 months of age ?

#3; L115 : From what age is it possible to detect macroscopically melanized SNpc in mice (in your model) ? Is it possible to measure a quantitative parameter of this macroscopic detection, such as CNR (contrast to noise ratio) ? Did you perform histological analyses on these mice ? It could be very interesting to know the exact origin of the hypersignal observed.

N=2 per group is too small to perform any correlation between histological results and MRI, but it could be very interesting, especially to validate your in vivo mouse model and to design further in vivo studies.

#4; L228 : At what age do you consider that your tgNM mouse model exhibits similar symptoms as prodromal/early PD stages? Pubertal? Young ?

#5; L289 : Can you precise the median survival of wt littermates?

#6; L325 : If the signal detected by MRI comes from iron, you should observe hypointensities on T2 and T2*w MRI too.

#Fig 1; What is the age of the mice on the MRI ? 12 months-old ? A better alignment between unstained macroscopic view and MRI slices of mice would be appreciated, as well as a colourbar for MRI images.

#Fig 2; Why are you using 2 different immunostainings for the WT and TgNM to assess NM accumulation?

#Fig 5F; Can you check your x-axis ?

#Fig 8G; What is the meaning of the dotted lines (black and grey) ?

To strengthen the discussion

- You should mention in perspectives correlation between in vivo MRI and histological results.
- As it is about a new mouse model, is there any endpoint that we should consider to design studies with this mouse model?

Supplementary Materials, L50

- Can you avoid abbreviations for non-MRI experts ?
- Did you acquire any scout images before T1 NM sensitive acquisition ?
- Can you precise the bandwidth that you used ?
- Why did you use images with and without fat saturation ? What are the parameters used for the fat saturation (position, thickness, ...) ?
- On the figure 1, did you show images with or without fat saturation or a combination of both the images ?
- Did you perform any post-processing step ? On your MRI, the signal seems to be quite homogeneous whereas you used a receiver surface coil.

POINT-BY-POINT RESPONSE TO REVIEWER COMMENTS

Reviewer #1 (Remarks to the Author):

In this study, Laguna and colleagues extended their previous work on the important connection between neuromelanin (NM) and neurodegeneration. Neuromelanin is absent in rodents and this is considered to be major limitation in modeling Parkinson's disease, where neuromelanin-rich neurons seem to preferentially degenerate. Here, the authors report the development of a tissue-specific transgenic mouse model based on the TH promoter-regulated expression of the enzyme tyrosinase, a key player in neuromelanin synthesis. The manuscript is essentially descriptive, but provides an important description of a model that may prove to be very useful for the field of PD research, given the existing limitations of the models available. With a few adjustments, I believe the study will be of interest for the broad scientific community.

Specific comments

As the authors mention, not all aged people develop PD/neurodegeneration. Presumably, they all have NM. The explanation that all humans would have incipient degeneration seems weak, as age-associated frailty can arise for many reasons besides neurodegeneration. An associated question is why would all mice show signs of disease?

It has been consistently demonstrated that there is a loss of catecholaminergic cells in the human brain with age (Beach et al., 2007; Mann and Yates, 1979; Xing et al., 2018). While this age-associated neurodegeneration may be due to multiple vulnerability factors endorsed by this specific cell type (i.e. long and diffuse axons, pacemaker activity and high metabolic demand), here we report that modelling one of them (i.e. NM accumulation) in rodents induces neuronal dysfunction and degeneration. It is also important to note that while all humans accumulate NM, not all humans show the same degree of intraneuronal NM accumulation. NM levels inside neurons vary with age and between individuals. In our previous study, we showed that PD and incidental LB disease (ILBD) cases (i.e. clinically healthy individuals exhibiting LB pathology considered to represent early, presymptomatic stages of PD) show increased intracellular NM levels compared to age-matched controls. The reasons for which these individuals show increased NM production and/or accumulation during their lifespan is still an unresolved question which might potentially be explained by exposure to different environmental factors or individual genetic differences. These individual differences may modify the risk of developing PD in humans. The case for mice differs substantially from that of humans in that mice used in this study are experimental models bred for research purposes in order to manifest consistent phenotypes. Littermates have a uniform genetic background (i.e. they were backcrossed for 8-10 generations using C57BL/6J mice prior to experimental assessment) and are kept under an exhaustive environmental control by the animal facility. Under these conditions, the levels of NM accumulated by mice are consistently reproduced and above a certain age the animals reach the pathogenic threshold set for neurodegeneration.

And why would there be NM in humans and not in rodents? Perhaps the authors can discuss this.

Different hypotheses have been postulated for the difference in pigmentation between different species. One potential explanation would be related to differences in lifespan, as NM only appears visible at the age of 3 years old in humans while median lifespan in rodents is around 2 years. Other explanations would be different metabolism rates and/or higher production of

catecholamines in humans than in mice, which consequently leads to different levels of oxidizing agents and NM accumulation.

What is the prevalence of eNM granules in the tg animals vs normal and PD brains? Can the authors include this comparison?

As suggested by the reviewer, we have compared the prevalence of eNM granules in tgNM mice and control and PD human postmortem brains. To allow such comparison across species, we have calculated a ratio between the number of eNM granules counted per SN section immunostained for TH divided by the area of the SN analyzed. In the graph below, the ratio for the three conditions is compared and shows that the prevalence of eNM granules in tgNM mice is higher than that in control brains and comparable to the one observed in PD brains. Although we agree with the reviewer that this data is of relevance, we are preparing a specific manuscript comparing different aspects of the neuropathology observed in tgNM mice with human brains and we would thus prefer not to include this data in the current manuscript.

Figure. Increased prevalence of extracellular neuromelanin (eNM) granules in human PD brains and tgNM mouse brains compared to human control brains. * $p \leq 0.05$ compared with control using a Kruskal-Wallis test with Dunn's multiple comparisons. Box plots: median, min-max values and individual dots for each animal.

Table S1 relates to data on Fig 2. What age is described in Table S1? It seems that in older animals A6 may have more NM. Is this significant?

Table S1 shows a qualitative assessment of brain pigmentation in animals at an adult age (i.e. 12 months). The different degrees of pigmentation correspond to distinct levels of NM accumulation visually depicted in the NM brain map shown in Fig. 2A. For simplification, only three degrees of pigmentation were established for the qualitative assessment (+, ++ and +++), so the most pigmented areas A1 to A10 were assigned to the high degree of NM accumulation, while the more rostral areas A10dc to A16 were assigned to the middle and low degree of pigmentation. As the reviewer points out, the precise age that was qualitatively assessed in Table S1 was not specified, so we have included this information in the table legend. Fig. 2B shows a quantitative assessment by optical densitometry of intracellular NM density at three

different ages throughout mice lifespan (young, adult and old). In this quantification, A6 shows statistically significant higher levels of NM accumulation compared to the other highly pigmented areas: (i) A9 and A10 in young (ii) A2, A9 and A10 in adult and (iii) A2 and A10 in old tgNM mice, as described in the figure legend.

Can the authors further characterize the LB-like inclusions in Fig 4 in order to support they are indeed LB-like? What about pS129 a-syn? Fibrillar a-syn antibodies? Ubiquitin? Recent LB-markers including mitochondrial remnants?

Without detailed characterization the authors should not refer to the structures observed as LBs (e.g. in line 180).

We referred to these structures as LB-like inclusions as they are defined by two LB-markers like the reactivity for p62 and alpha-synuclein, in half of the cases. As suggested by the reviewer, we have now further characterized these inclusions with additional LB-markers and, as shown in Figure S4E, all the p62-positive inclusions formed in tgNM mice are immunopositive for Ubiquitin. We did also try staining for phospho-synuclein (S129) using #ab51253 antibody and for synuclein filament using #ab209538 antibody but did not get any positive staining, despite the antibodies gave a positive staining in a brain section from an AAV-Synuclein injected rat. We also tried a recently described mitochondrial LB-marker like VDAC1 (#ab15895) as reported in Shahmoradian et al (Nature Neuroscience 2019; PMID: 31235907) with negative results. We have corrected the term in line 180 as indicated by the reviewer to read: *“The number of both LB-like inclusions and MB was less prominent in old tgNM,…”*.

Do the animals show SAA-positivity in CSF/skin/plasma?

We are currently setting up the SAA technique to detect positivity in samples derived from animal tissues and biofluids, but unfortunately this is something we cannot currently address.

Formation of LBs in human brain is not limited to melanized neurons (as cortical brain areas in DLB). The authors should consider this as it suggests NM is not necessary for LB formation.

LB in PD postmortem tissue are classified according to two different classes: (i) Classical brainstem LB in melanized neurons (i.e., eosinophilic cytoplasmic inclusions with a dense core surrounded by a pale halo of radiating fibrils); (ii) Cortical LB in non-melanized neurons (i.e., appear as irregular structures with rounded, angular, or reniform shapes without an obvious halo). Classical LB in brainstem melanized neurons are the first ones appearing in the course of the disease as postulated by Braak (Braak stages 1-3) while cortical LB appear only in later stages of the disease (Braak 4-5). Additionally, only LB-bearing melanized neurons degenerate in PD, while no neurodegeneration has been reported in LB-bearing non-melanized cortical neurons.

Here, we report that NM accumulation results in the formation of inclusions with LB-markers in rodents. So, in addition to other factors, NM accumulation in humans might trigger or contribute to the brainstem classical LB formation, while in later stages of the disease, other factors might as well induce cortical LB formation. In conclusion, we do not claim that NM is necessary for LB formation, but our data suggests that it might contribute to classical LB formation in melanized neurons by participating in the nucleation of the typical round classical LB structures surrounded by NM granules.

PD is more prevalent in males than in females. Did the authors observe gender differences? I suspect not, but perhaps the authors can show data to clarify this issue.

To reduce the number of animals used for the study, age variable was prioritized over sex variable, the latter not being included in the experimental design. Male and female mice were evenly assigned to all experimental groups to avoid sex bias. We did not observe remarkable differences between the two sexes, though no post hoc sex analysis was performed because of low sample size.

Reviewer #2 (Remarks to the Author):

Comments to the authors:

The manuscript “ Modelling human brain-wide pigmentation in rodents recapitulates age-related multisystem neurodegenerative deficits” by Laguna et al. generated a tissue-specific transgenic mouse (termed tgNM) that mimics the human age-dependent brain-wide distribution of neuromelanin (NM) within catecholaminergic regions and these animals display age-related neuronal dysfunction and degeneration affecting numerous brain circuits as well as peripheral organs, linked to motor and non-motor deficits, reminiscent of early neurodegenerative stages.

The group has validated the neuronal pigment induced by TYR overexpression in rodent catecholaminergic neurons closely resembles human NM. This supports the relevance of their approach to investigate the impact of human-like NM pigment on age-dependent neuronal function and viability in vivo.

The tgNM mouse model marks a significant milestone in the study of age-dependent NM accumulation’s potential influence on neuronal function and viability. This model offers a unique opportunity to address these questions with a context highly relevant to humans. These animals recapitulate the biological and temporal complexity of age-dependent neurodegeneration, affecting various brain neurotransmission systems and the body leading to motor and non-motor symptoms. The authors have found there were also the alpha-synuclein pathology and neuroinflammatory changes in this tgNM mouse model which is relevant to human non-diseased aged and PD brains. Those changes may start prior to evident neurodegeneration and be associated to the presence of incipient eNM released from early degenerating neurons. Even though not all the changes were dramatic, these animals could be used to combine with additional disease relevant genetic or environmental manipulations. Furthermore, these animals could also be used to test potential therapeutic strategies in a human disease state-informed context at different stages of the disease, which should facilitate the translation of findings to humans.

The work is really significance to the field. This model may open new research avenues in the field of brain aging and neurodegeneration.

However, there are some parts can be improved. So, the manuscript needs to have some major revisions to be published.

Major points:

1. For Statistics, the majority was t-test has been used to compare the WT and tgNM mice, so it is a pity that the age differences were not compared in most of the figures.

When parameters like NM density and LB-like inclusions are presented, only tgNM mice data is available and the comparison is done between the different age groups using a Kruskal-Wallis test, which allows to evaluate age differences. However, when tgNM and wt animals are analyzed at different ages, since all experiments have a relatively small number of mice and a

Gaussian distribution cannot be assumed considering sample size, we decided to apply only nonparametric tests. Thus, we decided to prioritize the genotype factor and compare the absolute values in tgNM and wt littermates at each given age with a t-test instead of representing data in tgNM over age (normalized to wt) and analyzing the age factor in a Kruskal Wallis test. Accordingly, we performed an analysis between genotypes at different ages and not an analysis of the aging factor per genotype.

2. In consistency of age groups, especially related to LC pathology, including figure 5 and S7, different ages group have been presented in different data sets.

Indeed, not all age groups were analyzed for every experimental approach. The large number of different techniques used in this study required different experimental setups and distinct processing steps. Consequently, the number of animals to be used increased significantly and for that reason we decided to optimize and reduce the number of animals following the principles of the 3Rs (Replacement, Reduction and Refinement) for performing more humane animal research. We processed brains at four different ages (pubertal, young, adult and old) for the characterization of neuropathology as this was the basis of the phenotype description (Fig. 5D, E, G, H and Fig. S7E, F, G). For some complex behavioral assessments that could not be done in-house and required shipment of mice to collaborating groups, only two age-groups were analyzed (adult and old) (Fig. 5A, B, C and Fig. S7B, C). Finally, to assess neurotransmitter levels by UPLC, additional groups of young and old mice were processed to have data on a wide age-range (Fig. 5F and Fig. S7D). We have noticed that there is a mislabeling in Fig. 5F where it should state “young” and “old” in both LC and PFC areas, which has already been corrected in the resubmitted version of the manuscript.

3. The discussion part is quite general, more summarized the previous work, not so much on the result from the paper.

We have worked on the discussion section to include specific points related to the results from the current paper.

4. ISH methods missing for figure S1C.

We apologize for this mistake and have included the corresponding methods for ISH in the revised version of the manuscript.

5. For DVC, TYR gene expression level is relatively low (figure S1A), but intracellular NM density is quite high (A2 in figure 2B), how to explain that?

We agree with the reviewer that this is an interesting observation, however, we suspect it is likely due to limitations of the technical approaches used for the two different experiments (i.e. TYR expression and NM density quantification). **The determination of TYR expression was performed by qPCR in an homogenate from bulk fresh-frozen dissections of the DVC area**, which comprises a relatively low ratio of catecholaminergic neurons (TH-positive) in relation to other cell types (e.g. cholinergic). Thus, TYR expression levels are expected to be low within this area compared to other areas that are mainly catecholaminergic, like A6, A9, A10 (Fig. S1A). However, the few catecholaminergic neurons present in the DVC show high **levels of intracellular NM density, which was quantified individually for each neuron (Fig. 2B)**.

Considering the limitation of the methodologies, we did not intend to correlate TYR expression with NM levels per brain region. TYR expression results (Fig. S1A) confirm that TYR expression increases with the number of catecholaminergic cells (TH-positive cells) in a specific brain region

as a confirmation of our transgenesis approach (TH-driven TYR expression). The intracellular NM quantification per neuron (Fig. 2B) indicates that the density of this pigment increases with age following a caudorostral pattern in the brain.

Minor points:

Line 210-211 “An in-depth analysis of PS evidenced significantly reduced numbers of bouts concomitant to a significant increase of their duration in both adult and old tgNM mice (Figure 5C)”. it is not correct since for the duration in adult was not significant.

We have corrected the manuscript: *“An in-depth analysis of PS evidenced significantly reduced numbers of bouts concomitant to a significant increase of their duration in old tgNM mice (Figure 5C), indicative of an irreversible dysregulation of PS ultradian rhythm”.*

2. The vocalization was recorded during the 6 min of the tail suspension test, why the number of mice were different between TST and vocalization?

We noted this difference in the vocalization pattern while performing the experiments for the tail suspension test (TST). Then we designed an additional behavioral assessment of the vocalization pattern in a new set of animals following the same experimental approach and that is the reason for the different number of animals in the two experiments.

3. Method part, line 203-204, “Data are expressed as optical density or absorbance defined by the logarithmic intensity of the light transmitted through the material using the formula: $-\log_{10}(\text{Striatum Intensity}/\text{Cortex Intensity})$ ”. Which data had been analyzed by the latter way? And where was the data for the old mice since it mentioned there was old mice in line 206-207?

As stated in the methods sections line 196, this data refers to *“The density of TH/DAT/VMAT2-positive fibers in the striatum, nucleus accumbens and olfactory tubercle was measured by densitometry in serial coronal sections covering the whole regions (4 sections/animal)”.* We have included data in Fig S4D showing the quantification for TH, DAT and VMAT2 immunostainings in old wt and tgNM mice.

4. There was no number of mice for Novel Object Recognition Test.

Indeed, this was not stated in the methods section and has now been added. Still, all genotypes, sample sizes and statistics are summarized in the Excel file named Data S1.

5. Volume and page information were missing for “Vila, M. (2019). Neuromelanin, aging, and neuronal vulnerability in Parkinson's disease. Movement disorders: official journal of the Movement Disorder Society.”

This has been corrected in the revised version of the manuscript.

6. TH antibody, 1:3000-1:20000 depending on the animal age, which method was used? The same for anti-ChAT.

The concentration determined here corresponds to the one used for TH- or ChAT-immunohistochemistry in the DVC. This has been specified in Table S2.

7. No information for ProCathepsin antibody

Cathepsin and Procathepsin were determined using the same antibody but separately identified by their different molecular weight. This has been clarified in Table S2.

8. Were LC3 I and II from the same gel and blotting?

We apologize but we realized we had wrongly identified LC3BI and LC3BII bands in our western blots. In the way the western blots were performed, the LC3BII band is very weak and cannot be quantified appropriately. We have thus modified figure SF6 to show only the band and quantification corresponding to LC3BI.

Points for figures:

Figure 1

There was no H&E and M-F staining for wt mice, the figure legends should be modified. This has been modified in the revised version of the manuscript.

Figure 3

A, wt adult, there must be some point(s) missing. This has been modified in the revised version of the manuscript. The number of adult wt animals used for this experiment was n=14. The raw data has been included in the *Source Data* excel file.

B, the Y axis of the figure is not correct, not second. This has been modified in the revised version of the manuscript.

E What could be the reason for TH positive cells of old mice increased instead of decreased? The graph depicts the absolute number of TH-positive cells counted in the stained sections for each animal and each time point. The immunostainings with sections from young, adult and old mice were done in different experiments since the animals were processed as they were sacrificed. Wild-type vs transgenic mice at each time point were always processed in parallel since the animals were littermates. Although the immunostaining protocol for TH is a routine staining in our lab and we always use the same reagents and incubation times, slight differences might be seen between sections processed in different experimental batches (i.e., slightly stronger or more faint staining). These differences in the staining might translate into differences in the absolute number of immunopositive cells counted in each batch, which could explain the higher number of cells counted in old mice. We could have shown the data as ratio wt vs tg for each time point but we always prefer to show the raw data as absolute number. The statistics have been done comparing wt vs tg at each time point and not across time points to avoid misinterpretations due to technical reasons.

F, what were the grey square and circle stand for? Was there no data for young mice?

The bar graphs in F represent the % of TH positive or negative cells within NM containing cells. The average % of TH positive or negative cells within NM containing cells is represented as the white or black parts of the bar graph, respectively. The grey square shows the percentage of TH+NM+ cells within NM+ cells for each animal as a way of showing the dispersion of the data for this parameter. The grey circle shows the percentage of TH-NM+ cells within NM+ cells for each animal as a way of showing the dispersion of the data for this parameter.

Data for young mice is not shown in F since we focused our analysis in adult and old time points where we have seen the differences in TH-positive cells reported in E.

The text in result parts showed that 3F could explain the result from E, but the comparison was different, E was the comparison between the WT and tgNM, F only show the age difference, how to conclude E from F?

The text the reviewer refers to reads “This decrease appeared to correspond to a TH phenotypic downregulation rather than cell loss, as indicated by the presence of dopaminergic NM-containing neurons immunonegative for TH (Figure 3F).” What we meant is that the presence of cells that had downregulated the phenotypic TH marker, both in adult and old mice (Fig 3F), was suggesting that the observed decrease in TH positive cells (Fig 3E) could not be due to cell loss but rather to this phenotypic downregulation, which has also been reported in human brains representing dysfunctional dopaminergic neurons at early stages of degeneration. The data shown in Fig 3G reporting no differences in the total number of dopaminergic neurons reinforces this hypothesis.

G, the Y axis is not correct, should be total NM containing neuron from the text, figure legend is also not correct.

Since NM is formed in all dopaminergic neurons, the number of NM-containing neurons represents a readout of the total number of dopaminergic neurons, irrespective of whether they have downregulated the phenotypic marker TH or not. The quantification criteria is explained in detail in the methods section. We have clarified this point in the text of the revised version of the manuscript.

Figure 4

C, What are the white circles and squares stand for?

The bar graphs in C represent the % of Synuclein-positive cytoplasmic inclusions. The black part of the graph represents the average % of Synuclein-positive inclusions in the different animals assessed while the white circle shows the percentage of Synuclein-positive inclusions for each individual animal as a way of showing the dispersion of the data for this parameter. On the other side, the white part of the bar represents the average % of Synuclein-negative inclusions in the different animals assessed while the white square shows the percentage of Synuclein-negative inclusions for each individual animal as a way of showing the dispersion of the data for this parameter.

Figure 5

A, The label of Y axis was confusing. We realize that the label might not be clear enough but for space constrictions we had to summarize in the Y label the information provided in the figure legend for A, which clearly explains the values that have been represented in the graph.

C, “s, seconds” should be added in the Y axis for the duration. This has been modified in the revised version of the manuscript.

ABC, no data for young mice, since LC related to non-motor symptom and changes already at Pebertal, behavioral tests should be done at young age to show the early onset of non-motor symptoms.

We completely agree with the reviewer in the importance of showing the early onset of non-motor symptoms related to LC function. However, the battery of motor and non-motor symptoms was initially designed only with adult and old mice because of the difficulties that manipulation young mice might represent sometimes, for example for the surgeries required to monitor sleep patterns. The initial behavioral assessments were performed prior to the histology results since the same animals were used for both purposes in order to reduce the number of animals required. After the histology results, we considered repeating some of the behavioral

tests in young mice and we actually did so when possible. Unfortunately, for the tests shown in Fig. 5A, B, C this was not technically possible.

F, no complete data for all four ages, figure legend was not precise.

We apologize since there was a mistake in the labelling of the X axis for this panel. The data shown is for LC and prefrontal cortex (PFC) NA levels in young and old tgNM and wt mice, not for the different four ages. This has been modified in the revised version of the manuscript.

Figure 6

C “cortex” in the result section, “PFC” in the figure legend while “Ctx” in the figure, should be consistent.

The terms have been unified for consistency and the term prefrontal cortex (PFC) is used in the text, figures and figure legends.

Figure 8

F, Ach was not mentioned in the result part.

Line 286 in the results section refers to the Ach levels shown in Figure 8F: “In addition, alterations in NA and Ach levels were also detected in vagal-innervated peripheral organs of tgNM mice like the heart and intestines (Figure 8F).”

Figure S1

A, there was significant difference between LC and DVC which was not indicated in the figure.

The statistical differences shown in the figure are between the different areas compared to the prefrontal cortex area taken as a reference area, as indicated in the figure legend and as shown in the *Source Data* file where the raw data and results from statistical tests are given. We have looked though again at the data and no significant difference was detected between the LC and the DVC in terms of tyrosinase expression.

B, It was mentioned in Figure legend that “In B; *p<0.05” but there was no “*” in the figure. How was the value of X and Y axis calculated?

As indicated in the figure legend, the linear regression between TYR expression and TH expression was positively correlated and this was statistically significant. We have modified the figure legend to better reflect this information. The value for the X and Y axis is calculated as -Delta CT for each gene compared to the housekeeping, as shown in the *Source Data* excel file.

What is the negative control of C?

The negative control for the in-situ hybridization shown in C is a sense probe made for tyrosinase so that there is no complementarity between the probe and the tyrosinase mRNA, in contrast with the antisense probe which is designed to be complementary and thus hybridize with the tyrosinase mRNA present in the tissue section. This information has been added to the methods section of the revised version of the manuscript.

Figure S4

A, were there individual dots missing for Th and Dat for Young mice? Were there only three samples for some of the group?

We have revised the graphs in Figure S4A and the raw data, now provided in the *Source Data* file, and there are no missing individual points. The minimum number of samples available for some groups was 4. The particular number of samples analyzed for each condition, ranging from n=4-11, is provided in the *Source Data* file.

C, the Y axis of the quantification is not proper.

We have modified the Y axis of the western blot quantification to read "TH or DAT protein levels (AU)".

S4C and S4D using different ages of mice

For consistency, we have included data in Fig S4D showing the quantification for TH, DAT and VMAT2 immunostainings in old wt and tgNM mice.

S4D, statistics was specifically motioned, was it different from all the others? The "*" was wrongly put in the figure. It will be good to have the image of the immunostaining.

We have corrected the misplaced "*" in the revised version of the manuscript and modified the figure legend since the statistics in D was the same as in A and C. As suggested by the reviewer, we have added a representative image of the immunostaining of striatal sections for TH, DAT and VMAT2 in wt and tgNM littermates in Figure SF4D.

Figure S5

no adult data even though figure 3A indicated that all the data had been collected for all the age groups.

The reviewer is right in that Figure S3A indicates that all data has been collected for all the age groups. For most of the various techniques used in this study the three time-points were analyzed. However, because of the large number of samples to process, for some of the techniques we prioritized the analysis of young and old mice, and that was the case for the UPLC analysis of neurotransmitter levels shown in Figure S5.

A, "SN" should be "SN-VTA"

This has been modified in the revised version of the manuscript.

D, the result of statistics was not mentioned.

This has been modified in the revised version of the manuscript and the details of the statistics are shown in the *Source Data* excel file.

Figure S6

The title of figure legend for S6 was wrong. This has been modified in the revised version of the manuscript.

A it seems that the quantification has been normalized to WT, hasn't it? As indicated in the methods section, band densitometry has been normalized to β -actin expression for each animal and then the relative expression compared to the average expression in wt mice has been calculated for each wt and tgNM sample.

B not correctly cited in the text and no result. The graphs in Fig S6B show the number of non-reactive Iba1+ cells in the SN and VTA regions and complement the quantification of reactive Iba1+ cells shown in Figure 4D. For that reason, both figures have been cited together in the text (line 187: TgNM mice also exhibited early inflammatory changes in the SNpc and VTA, including increased numbers of GFAP-positive astrocytic cells and reactive Iba1-positive microglial cells (Figure 4D and Figure S6B)). The result is that the number of non-reactive cells does not change and only the number of reactive cells is increased in tgNM mice compared to wt littermates.

Figure S7

B, It is a bit strange that the speed of old mice was higher than adult. We completely agree with the reviewer. Since the animals for the adult and old batches were different, and because they were assessed in different days, we cannot rule out technical differences in the experimental setup between both batches that go beyond our control. The statistical comparison has been done between wt and tgNM mice from the same age and assessed at the same time, and not between different ages.

F, What are the white circles and squares stand for?

As in Figure 4C, the bar graphs represent the % of Synuclein-positive cytoplasmic inclusions. The black part of the graph represents the average % of Synuclein-positive inclusions in the different animals assessed while the white circle shows the percentage of Synuclein-positive inclusions for each individual animal as a way of showing the dispersion of the data for this parameter. On the other side, the white part of the bar represents the average % of Synuclein-negative inclusions in the different animals assessed while the white square shows the percentage of synuclein-negative inclusions for each individual animal as a way of showing the dispersion of the data for this parameter.

Figure S8

B, The structures were indicated for UPLC right was not completely right. "serotonergic" should be added in the figure legend. This has been modified in the revised version of the manuscript.

Figure S9

A, spelling for histology was wrong.

D, "DNV" should be "DVC".

Both have been modified in the revised version of the manuscript.

Reviewer #3 (Remarks to the Author):

Comments to the authors:

This study is an original and innovative work. The authors evaluated and validated a new mouse model that better mimic human neuropathologies implying neuromelanin like in parkinson's disease. This work should open new ways to study neurodegenerative diseases as well as to test the efficacy of drug treatment. The authors combined different methods (behavioral testing, immunostaining, MRI, ...) to assess and explain the evolution of the brain pigmentation with age in this new mouse model.

General questions:

- In this study, did you use male or female mice? Can you explain your choice. If you used males and females, can you precise the male/female ratio?

In this study both male and female mice were used in all experiments and were distributed as evenly as possible to avoid sex bias. Since we used littermates as experimental groups (including both genotypes and both sexes), these followed approximately the expected 50% ratio of male and female. We did not include sex as an independent variable in the experimental design to reduce the number of animals used for the study. No post hoc sex analysis was performed because of low sample size.

- Did the tgNM mice present the same response to anesthesia (isoflurane) compared to wt ? In particular for microdialysis, peripheral functions assessment and MRI experiments ?

During in vivo procedures (i.e. microdialysis, peripheral functions assessment and MRI experiments), both tgNM and wt mice responded similarly to induction and maintenance doses of isoflurane anaesthesia. Similarly, both tgNM and wt mice showed similar recovery from anaesthesia.

More specific questions :

#1; L67 : I do not understand why you mentioned an “age-related loss of pigmented SNpc and LC neurons” in healthy aging brain whereas the 1st sentence in your introduction mentioned that “Humans progressively accumulate with age the dark-brown pigment neuromelanin (NM) within catecholaminergic brain nuclei”.

When stating that humans progressively accumulate NM within catecholaminergic cells we refer to a gradual increase of this pigment inside the neuronal cytoplasm, as neurons do not have the machinery to degrade this pigment. One hypothesis, supported by the findings presented in our study, is that the continuous intracellular build-up of NM within undegraded lysosomal-autophagic structures, until occupying most of the neuronal cytoplasm, may interfere with the normal cellular functions and ultimately lead to neurodegeneration. This would be in line with the observed decrease in the total number of catecholaminergic pigmented cells in the human brain with age. Indeed, extraneuronal granules of NM are also increased in aged individuals, possibly when being released in the extracellular space from dying melanized neurons.

#2; L107 : Why did you choose to perform this experiment at 12 months of age ?

The large number of different techniques used in this study required different experimental setups and distinct processing steps, thus increasing rapidly the number of animals required. For that reason, we decided to optimize and perform some of the experiments in only one age group to reduce the number of animals following the principles of the 3Rs (Replacement, Reduction and Refinement). In this experiment, we decided to choose 12 months of age as a middle age to measure TYR expression in tgNM mice.

#3; L115 : From what age is it possible to detect macroscopically melanized SNpc in mice (in your model) ? Is it possible to measure a quantitative parameter of this macroscopic detection, such as CNR (contrast to noise ratio)? Did you perform histological analyses on these mice ? It could be very interesting to know the exact origin of the hypersignal observed. N=2 per group is too small to perform any correlation between histological results and MRI, but it could be very interesting, especially to validate your in vivo mouse model and to design further in vivo studies.

We could only start detecting macroscopically melanized SNpc in our model at age 12 months by MRI, although when sectioning the brains, the melanized SNpc could already be seen faintly by eye in animals aged 3 months. Because the aim of this figure was to provide a qualitative illustration of the appearance of the pigment in tgNM mice compared to humans, we only used a very limited number of animals (N=2) and did not perform any quantification of the NM-MRI signal. We did not perform histological analyses on these mice since their brains were dissected after euthanasia and embedded in an agarose block to perform ex-vivo MRI imaging to confirm the origin of the hypersignal observed. We agree with the reviewer in that it would be interesting to confirm histologically in the same animals that the hyperintense signal observed corresponds to the pigmented cells that we visualize histologically in other littermates. Indeed, the usage of a non-invasive technique like MRI to monitor NM accumulation in vivo represents a useful tool for further in vivo studies aimed at testing for example different therapeutic approaches.

#4; L228 : At what age do you consider that your tgNM mouse model exhibits similar symptoms as prodromal/early PD stages? Pubertal? Young ?

We consider that our mice exhibit similar symptoms as prodromal/early PD at adult and old age, since we do not observe overt nigral neurodegeneration up to 20 months of age. However, we do see noradrenergic neurodegeneration already at a pubertal age, supporting that noradrenergic denervation might precede nigral degeneration as well in human PD. In line with this, noradrenergic denervation has indeed been observed in patients with idiopathic rapid eye movement (REM) sleep behavior disorder (RBD), which is considered a prodromal form of PD.

#5; L289 : Can you precise the median survival of wt littermates?

To reduce the total amount of animals used for the study we did not plan an independent experimental group to perform the survival analysis but instead, we followed the animals already assigned to the different experimental groups and annotated the occurring natural deaths. TgNM mice showed more natural deaths than wt littermates, thus reaching 50% survival (median survival; 20.7 months of age) before their experimental euthanasia, whereas wt littermates were euthanized for experimental purposes before reaching median survival. For this reason, an exact median survival age for wt littermates in this study could not be determined. Still, tg and wt littermates were in a C57BL/6J genetic background and these have been extensively studied in numerous survival analysis and normally show a median age survival of 24-28 months.

#6; L325: If the signal detected by MRI comes from iron, you should observe hypointensities on T2 and T2*w MRI too.

The reviewer is right in pointing that hypointensities should also be observed on T2 and T2*w MRI. It has been described that hyperintensity in T1w coincides better with the SN than hypointensity in T2w and that is why we decided to make high-resolution T1w images. According to the literature, this is likely due to the fact that the T1 shortening effect on MRI caused by iron is weaker and more localized than the blurring effect of the T2* effect in this area, especially at high field strength. Reference: Neuromelanin-related T2* contrast in postmortem human substantia nigra with 7T MRI. Scientific Reports | 6:32647 | DOI: 10.1038/srep32647

#Fig 1; What is the age of the mice on the MRI ? 12 months-old ? A better alignment between unstained macroscopic view and MRI slices of mice would be appreciated, as well as a colourbar for MRI images.

The age of the animals from the MRI images in Figure 1 is 16 months of age. It is more usual to present the quantitative maps of T1 and T2 in color with their colourbar, but the standardized criterion to display weighted images with gray scale. For that reason, the T1w images shown in Figure 1 are displayed with the grey scale.

#Fig 2; Why are you using 2 different immunostainings for the WT and TgNM to assess NM accumulation?

In Fig. 2A we show all catecholaminergic groups spanning the rostro-caudal extend of the non-pigmented wt mouse brain with an immunostaining of the catecholamine-synthesizing enzyme TH (first raw of images). Then we show the same catecholaminergic groups in tgNM mice, which accumulate NM, using Nissl staining (second raw of images) for a better visualization of the dark-brown pigment. The images shown in Fig. 2A are only for visualization purposes and qualitative assessment. For the exact quantification of NM accumulation in the most pigmented groups we used unstained slices to avoid any stain interference with the color/density of the pigment (Fig. 2B).

#Fig 5F; Can you check your x-axis ? This has been modified in the revised version of the manuscript.

#Fig 8G; What is the meaning of the dotted lines (black and grey)?

Dotted lines represent the 95% confidence interval for each genotype. This information was lacking and it has been added in the figure legend of the revised version. Grey lines represent wt mice while black lines represent tgNM mice as indicated in the figure legend.

To strengthen the discussion

- You should mention in perspectives correlation between in vivo MRI and histological results. This has been added to the revised version of the manuscript at the end of the discussion section.

- As it is about a new mouse model, is there any endpoint that we should consider to design studies with this mouse model?

As discussed in the results section, tgNM mice show a median survival of 20.7 months of age. At this age, no overt neurodegeneration was observed in the SNpc region while extensive degeneration was detected in the LC already from young ages. The extent of early neuropathological features in the SNpc, like extracellular NM granules, TH downregulation, p62-positive inclusions and neuroinflammation markers, could be used as endpoint markers in the design of future studies using this mouse model. Importantly, the motor and non-motor behavioral alterations observed, some from already early ages, are important outcome measures to consider.

Supplementary Materials, L50

-Can you avoid abbreviations for non-MRI experts? We apologize for this inconvenience, we have added more information in the methods section to describe the abbreviations used.

-Did you acquire any scout images before T1 NM sensitive acquisition ? Yes, indeed low-resolution T2-weighted fast spin-echo images were acquired before T1 NM sensitive images. This has been added to the methods section of the revised version of the manuscript.

-Can you precise the bandwidth that you used ?

For T1w images, the effective Spectral Bandwidth used was 73529.4 Hz.

-Why did you use images with and without fat saturation ? What are the parameters used for the fat saturation (position, thickness, ...) ?

Images with and without fat saturation were acquired in the in vivo MRI experiments to see if we had better NM contrast in the brain by suppressing the subcutaneous fat, which saturates the image a little because it is very intense in the T1w images. Fat suppression was thus used to check if the high signal from fat was obscuring NM contrast. For fat suppression, a frequency-selective 90 degree gauss512 pulse was globally applied to a frequency offset of -3.5 ppm relative to water (fat suppression bandwidth=1051.16Hz). Transverse magnetization produced by the pulse was suppressed with a gradient spoiler.

-On the figure 1, did you show images with or without fat saturation or a combination of both the images?

The images shown in Figure 1 correspond to images acquired ex-vivo without fat saturation. This has been added to the revised version of the manuscript.

-Did you perform any post-processing step? On your MRI, the signal seems to be quite homogeneous whereas you used a receiver surface coil.

No post-processing step was applied. The surface probe used to receive the signal is curved to adapt to the mouse head so that the signal from the brain is observed as quite homogeneous. For excitation, a volume probe was used that offers homogeneous excitation. The methods section has been modified to better explain these details.

REVIEWERS' COMMENTS

Reviewer #1 (Remarks to the Author):

The authors have addressed my comments appropriately, and I have no further comments at this stage.

Reviewer #2 (Remarks to the Author):

All my concerns have been addressed in the revisions. I am satisfied with it.

Reviewer #4 (Remarks to the Author):

Summary

This is a resubmitted paper. The authors describe a new mouse model in which neuromelanin is produced in catecholaminergic neurons by expression of human tyrosinase under control of the mouse tyrosine hydroxylase promoter. A number of different strategies are used to characterize the effects of transgene expression – including macroscopic and microscopic imaging, neuromelanin accumulation, behavior changes, increase of pigmented tyrosine hydroxylase immunonegative cells, intracellular inclusions, extracellular neuromelanin, autonomic dysfunction, and reduced life span – demonstrating similarities between the new model and expected human pathology. I have been asked to restrict my comments to whether the authors have satisfactorily addressed comments by previous reviewers.

Comments

- 1) Overall, the authors have done a satisfactory job in responding to past reviewer comments. However, some of the details are provided in the rebuttal but not also put into the revised manuscript or its supplements. I accept the statement that the figure comparing extracellular neuromelanin in post-mortem human and transgenic mouse tissue is the topic of a separate manuscript, but other details are excluded for no apparent reason. For example, the MRI details requested by Reviewer 3 are omitted from the paper. The fat suppression and bandwidth details should be put into the appropriate methods section and not just in the rebuttal.
- 2) The legend for Figure 1 should explicitly state that the MR images shown were acquired *ex vivo*.

POINT-BY-POINT RESPONSE TO REVIEWER COMMENTS

Reviewer #1 (Remarks to the Author):

None

Reviewer #2 (Remarks to the Author):

None

Reviewer #4 (Remarks to the Author):

1) "The MRI details requested by Reviewer 3 are omitted from the paper. The fat suppression and bandwidth details should be put into the appropriate methods section and not just in the rebuttal."

This is now provided in the Materials and Methods section, page 12

2) "The legend for Figure 1 should explicitly state that the MR images shown were acquired ex vivo."

This is now indicated in the Figure 1 legend.